# Oceanic response to changes in the WAIS and astronomical forcing during the MIS31 superinterglacial

Flavio Justino[1], Douglas Lindemann[1], Fred Kucharski[2], Aaron Wilson[3], David Bromwich[3], and Frode Stordal[4]

[1]Department of Agricultural Engineering, Universidade Federal de Vicosa, PH Rolfs, Vicosa, Brazil
[2]The Abdus Salam International Centre for Theoretical Physics, Trieste, Italy
[3]Polar Meteorology Group, Byrd Polar and Climate Research Center, The Ohio State University, Columbus, OH, USA
[4]University of Oslo,Department of Geosciences, Forskningsparken Gaustadalleen, Oslo, Norway

*Correspondence to:* Flavio Justino (fjustino@ufv.br)

**Abstract.** Marine Isotope Stage 31 (MIS31, between 1085 ka and 1055 ka) was characterized by higher extratropical air temperatures and a substantial recession of polar glaciers compared to today. Paleoreconstructions and modeling efforts have increased the understanding of MIS31 interval, but questions remain regarding the role of the Atlantic and Pacific Oceans in modifying climate anomalies associated with the variations in Earth's orbital parameters. Based on multi-century coupled climate simulations, it is shown that under the astronomical configuration of the MIS31 and forced by modified West Antarctic Ice Sheet (WAIS) topography, there exists a substantial increase in the thermohaline flux and northward oceanic heat transport (OHT) in the Pacific Ocean. These changes are driven by anomalous wind-driven circulation and increased surface salinity in concert with stronger meridional overturning circulation (MOC), resulting in greater northward OHT that contributes up to 85% of the global OHT anomalies, adding to an overall reduction in sea-ice in the Northern Hemisphere (NH) due to Earth's astronomical configuration at the time. Relative contribution of the Atlantic Ocean to global anomalies are minor compared to those related to the Pacific insofar as the OHT and MOC are concerned. However, sea-ice changes in both hemispheres are remarkable. In the Southern Hemisphere (SH) changes are highlighted by decreased (increased) cover in Ross (Weddell) Sea, whereas in the Northern Hemisphere reduction is largely noted in all latitudes.

## 1 Introduction

*During the last decades substantial research efforts have aimed to investigate past climates from paleoreconstruction and climate modeling experiments, focussing on disentangling the influence of dominant climate forcing, such as orbital configuration (Yin (2013); Erb et al. (2015)), and/or particular processes including ENSO and ice sheet variability (Russon et al. (2011); DeConto et al. (2012)). However, many issues are still required to be fully addressed concerning the nature of long-term changes associated with the air-sea coupling due to its impact on atmospheric and oceanic variability (Knutti et al. (2004); Bush and Philander (1998)). This twofold interaction, implies that the atmosphere affects the sea surface conditions through modification of the oceanic heat fluxes which feed back to the lower tropospheric atmospheric flow (Timmermann et al. (1998)).*

*The global climate response to these processes is governed by a complex interaction relying on processes not only occurring at the air-sea interface but also in sub-surface oceanic layers where a substantial amount of heat is stored (Meehl et al. (2011); Yin and Berger (2012); Ganachaud and Wunsch (2003)). Therefore, the use of fully coupled models is essential to reproduce large-scale climatic features such as operating in glacial and interglacial climate (e.g Erb et al. (2015); ?). In particular, when the climate response to potential changes in the Meridional Overturning Circulation (MOC) and the Oceanic Heat Transport (OHT) are of concern (Shin et al. (2003)).*

*The present study adds to previous analyses conducted with simplified climate model who applied atmospheric general circulation model coupled to slab mixed-layer ocean model. Though valid, this modeling approach reduces ocean-atmosphere feedbacks that are crucial for the reorganization of atmospheric flow in long timescales. It has been emphasized by coupled modeling studies that oceanic dynamical changes related to orbital fluctuations is the primary forcing in determining large-scale atmospheric flow patterns (Erb et al. (2015); Tomas et al. (2016)).*

*Indeed, increased OHT from the Pacific into the Arctic associated with changes in Antarctic ice volume has been argued to affect the Beringian climate during interglacial epochs (Coletti et al. (2015)). Moreover, it has been recognized that increased OHT has often been claimed to maintaining warm high-latitude surface temperatures in many intervals of the geologic past (Comeau et al. (2016)). This approach is explored here by using a coupled model.*

*An interesting test case to explore these climate feedbacks is the Marine Isotope Stage 31 interval (MIS31) which occurred at ∼ 1080 ka BP (Lisiecki and Raymo (2005)). This epoch has been chracterized by warmer global air temperatures and substantial melting of polar glaciers compared to today (Melles et al. (2012); Wet et al. (2016)). However, paleoreconstructions and modeling results disagree with respect to the North Hemisphere warming during the MIS31 suggesting the need for a better understanding of this interglacial and other warmer climates (Coletti et al. (2015); Melles et al. (2012)) as well. At large, these differences may arise from limited constrained paleoreconstructions with improper climate model boundary conditions. Moreover, modeling interglacial stages requires changes in internal and external forcing involving modifications of the ice sheet topography (Pollard and DeConto (2009); Melles et al. (2012)), atmospheric $CO_2$ concentration (DeConto and Pollard (2003)), and the planetary astronomical configuration (Erb et al. (2015); Yin (2013)).*

*To gain insight on the matter, analyses have focused on the climate response to individual drivers of the interglacial climates (Knorr and Lohmann (2014); Yin and Berger (2012); Pollard and DeConto (2009), Villa et al. (2008)). Among other effects, insolation have been shown to play the dominant role in defining high-northern latitude temperature and sea-ice (Yin and Berger (2012)). The longitude of the perihelion (precession) is also found to lead changes in the equatorial Pacific seasonal cycle (Erb et al. (2015)). Meanwhile, past fluctuations in atmospheric $CO_2$ concentration have been claimed to induce long-term surface and deep-water temperature trends (Knorr and Lohmann (2014)).*

*Accordingly, this study aims to disentangle the individual contributions of the WAIS and the astronomical configuration during the MIS31 climate. Mechanisms related to the combined effects of these forcing and associated with the inter-hemispheric coupling, including the potential role of the OHT, wind-driven and thermohaline changes, as regulator of the MIS31 anomalous climate are explored. Because oceanic dynamical changes during interglacial intervals are crucial for determining the large-scale atmospheric circulation and temperature distribution (Coletti et al. (2015)), answers to these issues are pursued by*

*employing the International Centre for Theoretical Physics - Coupled Global Climate Model (ICTP-CGCM) (Kucharski et al. (2015)). The astronomical forcing is assumed to represent 1072 ka based on the warmest summer month in lake El'gygytgyn reconstruction (Coletti et al. (2015); Melles et al. (2012)). Results provide insight on the air-sea exchange processes and large-scale ocean dynamics characteristic of that epoch.*

## 2   Coupled model and experimental design

The ICTP-CGCM control simulation (CTR) is run under present day orbital forcing for over 2000 years since proper evaluations of long-term ocean-atmosphere processes require statistical equilibrium representation of the climate state, particularly for paleoclimatic features in a coupled atmosphere-ocean model (Peltier and Solheim (2004)).

*The $CO_2$ concentration in our CTR climate is 325 ppm because it characterizes the CO2 concentration by the year 1950 which does not include the fast rate of increase in $CO_2$ due to human emission occured by the end of the 20th century. The ICTP-CGCM, consisting of the atmospheric global climate model "SPEEDY" version 41 (Kucharski et al. (2006)) coupled to the Nucleus for European Modelling of the Ocean (NEMO) model (Madec (2008)) with the OASIS3 coupler (Valcke (2013)), is used in this study. The atmospheric component runs at T30 horizontal resolution and there are eight levels in the vertical. The model includes physically-based parameterizations of large-scale condensation, shallow and deep convection, shortwave and longwave radiation, surface fluxes of momentum, heat and moisture, and vertical diffusion. NEMO is a primitive equation z-level ocean model based on the hydrostatic and Boussinesq approximations. This version applies a horizontal resolution of $2°$ and a tropical refinement to 0.5°. The ocean component has 31 vertical levels with layer thicknesses ranging from 10 m at the surface to 500 m at the ocean bottom (16 levels in the upper 200 m). Additional details of the ICTP-CGCM are described by Justino et al. (2015) and Kucharski et al. (2015).*

### 2.1   Model performance of the CTR climate

To evaluate the reliability of the coupled model to represent the present day climate (control run), Fig. 1 shows SST differences between the CTR run and the NOAA Optimum Interpolation (OI) Sea Surface Temperature V2 (NOAA-OI-SST-V2) (Reynolds et al. (2002)) and sea-ice area based on the Hadley Centre Sea Ice and Sea Surface Temperature data set (HadISST) (Rayner et al. (2003)). The control run has been run for 1000 years and CTR climatology is based on the last 100 years. The modeled evaporation minus precipitation (E - P) flux is compared to the Interim Reanalysis from the European Centre for Medium-Range Weather Forecasts (ECMWF) ERA-Interim (ERAI) (Dee et al. (2011)). It has to be mentioned that Kucharski et al. (2015) has provided detailed analyses of the present day climate simulated by the ICTP-CGCM.

*Comparison between the ICTP-CGCM and NOAA-OI-SST-V2/HadISST for the annual SST pattern (Fig. 1a) shows differences in the extratropical ocean where the ICTP-CGCM is colder than NOAA-OI-SST-V2 that can be related to differences in the lower tropospheric flow. In fact, the zonal wind over the NH strom track region in ICTP-CGCM as well as over the SH polar jet are weaker than in ERAI up to ± 4m/s. In the NH during the summer season, this will be associated with reduced*

*temperature advection from Asia and North America onto the North Atlantic and Pacific leading to lower ICTP-CGCM SSTs. However, it should be stressed that overall SST differences are in the range of ± 2°C.*

*Analysis of E - P flux demonstrates that our coupled model is able to reasonably reproduce the main characteristics of the ERAI E - P flux (Fig. 1b), but the zonal averages reveal that the ICTP-CGCM is wetter than the ERAI in the equatorial belt and SH mid-latitudes (not shown). However, differences are less than 1 mm day$^{-1}$. This implies that the E - P flux associated with the Intertropical Convergence Zone (ITCZ) needs improvements in order to better reproduce equatorial climate dynamics including decreased precipitation in the Pacific Warm Pool and over the southern part of the South Atlantic. Nevertheless, this is a recurrent feature in other CGCMs (Jia-Jin (2007))*

*Comparison of the ICTP-CGCM sea-ice area with estimates from the Hadley Centre counterpart shows that the ICTP-CGCM does a reasonable job in both hemispheres for December-January-February (DJF) and June-July-August (JJA) (Table 1). Sea-ice area is computed as the total area covered by ice, which corresponds to sum of the area of each cell multiplied by the fractional concentration for that cell. Insofar as annual mean conditions are concerned (Fig. 1c,d), limitations are evident as ICTP-CGCM is dominated by higher sea-ice concentration than delivered by the Hadley Center in most of the NH, in particular in the Russian Arctic (Fig. 1c). In the SH our model is characterized in the Atlantic polar region by lower concentration of ice but higher concentration in the sea-ice edge in extratropical latitudes around 60°S (Fig. 1d), and at Ross Sea. The performance of the ICTP-CGCM to reproduce the annual cycle of sea-ice thickness in the Weddell Sea exhibits higher amplitude seasonal cycle as compared to HadISST, with thinner sea-ice in summer (not shown).*

*Several investigations demonstrated that extra-tropical SST and sea-ice are currently among the largest limitations of Earth climate modeling. Based on a CORE model intercomparison (Griffies et al. (2009)), we have found that the ICTP-CGCM biases are in the lower range as compared to other models. In this intercomparison with driven ocean only simulations it was demonstrated that even in this idealized scenario generally models appear to have large biases in all fields (such as SST, SSS, sea-ice, zonal velocity in Equatorial Pacific subsurface, see Griffies et al. (2009) Figs. 7 and 8). In particular, the models' AMOCs (their Fig. 23) show substantial spread, but Kiel-ORCA performs very well.*

*The ICTP-CGCM, which is a coupled ocean-atmosphere model that applies the same Kiel-ORCA ocean component allows to draw confidence that our ocean component is among better models of the CORE model intercomparison. This is supported by a recent publication by Kucharski et al. (2015). As discussed further in our study, the ICTP-CGCM AMOC exhibits values that closely match observation such as in Kanzow et al. (2010); Ferrari and Ferreira (2011); Talley et al. (2003), as well as compared to higher resolution models (Stepanov and Haines (2014)). Consequently, a fair representation of the AMOC should lead to proper oceanic heat transport (OHT) estimates under present day conditions because the majority of the OHT is driven by the AMOC. Moreover, ICTP-CGCM is run in a reasonable resolution for a global model in particular in the tropics where most of the OHT is transported. This is discussed in more details ahead.  Design of the sensitive experiments*

To evaluate the climate impact of changes in the WAIS topography and the astronomical forcing during the MIS31 inter-glacial, 3 additional sensitivity experiments have been conducted and the analyses are carried out for the last 100 years of 1000 years-long simulations (Table 1 supp. material):

1. TOPO - applies the WAIS topography as proposed by previous studies (Pollard and DeConto (2009); Justino et al. (2015));

2. AST - conducted with astronomical configuration characteristic of the 1072 ka (Berger (1978); Coletti et al. (2015));

3. MIS31 - the combined effect of the forcing described in TOPO and AST.

*In all sensitivity experiments the $CO_2$ concentration is set to 325 ppm. For the MIS31 interval this is reasonable based on boron isotopes in planktonic foraminifera shells (Honisch et al. (2009)). The $CO_2$ concentration during the MIS31 could vary between 300 and 350 ppm due to propagated error of the individual pH, SST, salinity, and alkalinity in the reconstructions. This variation in the amount of atmospheric $CO_2$ may lead to an overestimation in the NH warming as simulated in our study. Changes in $CO_2$ by about +50 ppm may be associated with +0.3K change in globally averaged surface temperature. But, this alteration in temperature is within the uncertainties of the climate sensitivity (Bindoff et al. (2013)). The CH4 (800 ppb, Loulergue et al. (2008)) and N2O (288 ppb, Schilt et al. (2010)) concentrations are similar to Coletti et al. (2015).*

*Though the Greenland Ice Sheet (GIS) may have been reduced as compared to present day, Coletti et al. (2015) shows that MIS5e is warmer than MIS11 and MIS31. Since it is known that GIS still existed but was slightly smaller during MIS5e than currently, it can be reasonably assumed that it was much similar during MIS31 as compared to today. Therefore, the GIS in ICTP-CGCM reflects present day conditions. Our simulation does not include changes in oceanic gateways, because there is no conclusive global land-sea mask reconstruction for the MIS31 interval. The WAIS topography has been modified, but no changes in sea level have been applied in our modeling experiment. However, the modified WAIS reflects sea water albedo in the sensitivity runs.*

*Changes in the initial salinity field in response to the WAIS collapse have not been included. Aiken (2008) demonstrated limited response of the climate system to the freshening implied by Antarctic sea-ice melt, even in the presence of adding much larger freshwater forcing of approximately 0.4 Sv ($10^6 m^3 s^{-1}$). Moreover, Vaughan and Arthern (2007) argued that an outflow rate associated with WAIS melting is not realistically attainable, making it difficult to implement in a rose experiment. Insofar as the WAIS collapse is concerned this study focuses on analyzing the climate response to mechanical changes in orography.*

## 3  Climate response to MIS31 forcing

*The WAIS collapse*

Under present-day conditions, katabatic winds flowing offshore from the continent over the Weddell Sea contribute for maintaining cold air over the sea-ice edge (Mathiot et al. (2010)). Modeled Weddell Sea warming in the TOPO simulation is

related to weaker katabatic winds and reduced continental cold air advection (Justino et al. (2015)) due to a collapsed WAIS. In conditions of reduced sea-ice thickness there is an increase in the heat flux from the ocean to the atmosphere further increasing the convective mixing warming the overlying atmosphere. Higher temperatures in the Ross Sea in TOPO is supported by the Ocean Drilling Program (ODP) site 1165 and by the marine glacial record of the AND-1B sediment core in the Ross Ice Shelf (Naish et al. (2009)).

*The WAIS collapse also leads to the North Atlantic cooling in response to the slowdown of the AMOC (as discussed later). Temperature anomalies of opposite sign between the North and South Atlantic have been a recurrent feature in the Earth climate related to the North Atlantic freshening (Knutti et al. (2004)) or modification of the SH wind patterns and subsequently wind-driven circulation (Speich et al. (2007)).* Previous work (Justino et al. (2015)) using a simplified low resolution ocean model ($3° \times 3°$) has shown that the incorporation of a modified WAIS topography characteristic of the MIS31 interval, results in generally warmer global surface temperatures with enhanced positive anomalies between 50-70°S. It should be noted that the warming delivered by the current ICTP-CGCM TOPO simulation is substantially smaller as compared to previously found (Justino et al. (2015)) (Fig. 2a). Lower surface temperature anomalies noted in the Ross and Weddell Seas only extend out to 40°S. Seasonal Seaice changes (DJF and JJA) between TOPO and CTR shows increase in area in both hemisphere with larger changes in the NH which are not statistically significant. It may be pointed out that the effect of WAIS changes amplify the effect of the orbital forcing as discussed later (Table 1). Annually averaged, however, sea-ice concentration changes due to WAIS collapse have to be considered in the Labrador and Nordic seas and most of the SH (Fig. 2d). This indicates that the Autumm and Spring season dictates changes of annual sea-ice conditions in the WAIS simulation.

There are several factors related to differences in global temperature anomalies between the current study and Justino et al. (2015), which also evaluate the climate response to the WAIS collapse. In fact, the previous model results show much weaker SH westerly flow leading to warmer SSTs across the high latitudes of the SH compared to the present study (not shown). The previous study also reflects weaker teleconnections between the tropical and extratropical regions related to the El Nino-Southern Oscillation (ENSO) (Severijns and Hazeleger (2010)). NEMO (present model) and CLIO (previous model) are characterized by drastically modified ENSO related-tropical variability in terms of variance and magnitude (Severijns and Hazeleger (2010); Park et al. (2009)). The NEMO ocean model used in the present study can properly simulate the global oceanic features as it resolves convective and mesoscale processes in the mixed layer and themocline related to the ENSO.

It has long been recognized that the effect of the air-sea coupling by the Ekman layer for the surface climate is remarkable. For instance, CGCMs driven by a lower resolution oceanic component are very limited in their ability to reproduce the wind-driven upwelling, and therefore are warmer than those models running with higher resolution. More importantly, low resolution ocean-atmosphere models struggle to reproduce the OHT. In this line, it has to be mentioned that SPEEDO simulates weaker Atlantic Overturning Circulation (NADW = 8 Sv), and therefore allows for larger storage of heat in the Southern Hemisphere due to North Hemisphere heat piracy assumption (Broecker (1998)). The NADW in the ICTP-CGCM matches observations (22 Sv) closely.

*The AST forcing*

Turning to the impact of astronomical changes on global surface temperatures (AST minus CTR), warming is evident in the northeastern Pacific and Atlantic Oceans (Fig. 2b). The orbital forcing during the MIS31 interval are characterized as high obliquity and eccentricity enhance boreal summer insolation. Downward solar radiation differences at the top of the atmosphere between AST and CTR reach values of up to 50 W m$^{-2}$ at 60°N (not shown). In fact, increased heat in the oceanic surface layer during the summer months hinders the winter cooling which over extratropical latitudes hampers sea-ice cover (Yin and Berger (2012); Alexander et al. (1999)). Thus, vigorous oceanic heat exchange leads to higher near-surface air temperatures compared to the CTR run. It has to be highlighted that seasonal changes project onto annual conditions due to the remnant insolation effect, that is stronger during the NH summertime (Yin and Berger (2012)). Indeed, the NH (SH) warming (cooling) is primarily associated with intensified (weakened) summer insolation that is dominant in the polar and subtropical regions. In addition to reduced insolation in the SH, stronger southeast trade winds and westerlies (Fig. 2d) lead to lower surface temperatures related to stronger equatorial upwelling and modified Ekman dynamics (McCreary and Lu (1194)). The wind-evaporation-SST feedback also plays a role due to modification in the latent heat through evaporation (Wang et al. (1999)).

Elsewhere, the atmospheric circulation and the heat exchanges due to air-sea interactions determine annual mean conditions. The incorporation of the astronomical forcing also delivers anomalous surface temperature patterns in such a way that the Atlantic Ocean anomalies resemble present-day conditions under the positive phase of the Atlantic Multidecadal Oscillation (AMO) (Delworth and Mann (2000)). Surface temperature anomalies in the North Pacific on the other hand, depict the warm phase of the Pacific Decadal Oscillation (Zhang et al. (1997)). *These climate features (AMO and PDO), therefore, may be characteristic of a global climate governed by an excess of heat in the NH as occurred in the MIS31 interval. Thus, the 20th century climate which experienced larger changes of NH temperatures as compared to the SH counterparts (Kang et al. (2015)) will also display the AMO and PDO as dominant anomalies. In today's climate the NH warming arises in part because of northward cross-equatorial ocean heat transport (Kang et al. (2015)), however, the heat transport has to be intensified under distinct external forcing such as for interglacial climates. This will be discussed further.*

*Table 1 and Figure 2e demonstrated that the inclusion of orbital forcing leads to decreased (increased) in sea-ice area in the NH (SH) with statistically significant changes in the NH. Though, changes in the SH during JJA experience larger magnitude. However, Figure 2e shows that when annual average is taken most changes in the SH are significant and that the ORB anomalous sea-ice pattern opposes the TOPO response in particular in the Weddell and Bellinghausen Seas. Surface*

*climate response to joined AST and WAIS forcing*

The global climate response due to combined effect of changing WAIS topography and astronomical forcing (MIS31 simulation) is primarily a result of changes in the latter forcing, as Fig. 2c shows a similar surface temperature anomaly pattern as Fig. 2b. Nevertheless, the combined forcing appear to be not linear in the vicinity of Antarctica (supplementary material Fig. 2). Intensified warming is shown in the Ross Sea (the result of warmer surface temperatures in TOPO and AST) but reduced

cooling in the Weddell Sea, where the absence of the WAIS topography in the joint effect reduces the strong cooling associated with changes in the astronomical forcing. Comparison between the MIS31 and the AST runs can be indirectly used to further identify the effect of the WAIS topography in sea-ice changes.

The surface temperature anomalous patterns in the NH are also associated with modify air-sea coupling in particular reduced Ekman drift and reduced evaporative cooling in consonance with the MIS31 orbital forcing. Indeed, stronger mid-latitude and polar westerlies over the Kuroshio/Oyashio region and weaker northeast trade winds over the central-eastern Pacific (not shown) result in higher SST in the respective regions. Changes in surface temperature and winds generate sea-ice anomalies (Table 1, Fig. 2). Modification of the WAIS topography is associated with changes in sea-ice area particularly in the Atlantic Ocean. Changes in the astronomical forcing on the other hand are responsible for climate anomalies in a global perspective. These results cast uncertainty on previous studies based on in situ reconstructions that assume overall warming and sea-ice free conditions in the Southern Ocean as compared to the present-day climate (Scherer et al. (2008)).

*Differences between MIS31 and AST usefully demonstrate that the substantial reduction of sea-ice cover in the Ross Sea and in some extent changes in Weddell Sea are substantially affected by the WAIS collapse (Supplementary Fig. 1c). Specifically, the MIS31 simulation is warmer in the Weddell and Ross Seas by up to 1.5°C with respect to AST, which is accompanied by reduce sea-ice cover by about 10%. In fact, the individual influence of the WAIS (ORB) to MIS31 is more evident in the Bellinghausen (Weddell) Sea (Fig. 2 e,f). In the NH, the WAIS forcing clearly dumps the effect of the orbital forcing in sea-ice changes.*

*The sensitivity experiments demonstrate that warmer surface temperatures and reduced sea-ice are only simulated in the Ross Sea region, in agreement with the Cape Roberts Project-1 results and data from the Antarctic Geological Drilling project (ANDRILL) (Naish et al. (2009)) (Fig. 2). In fact, outside of the Ross Sea, Antarctic sea-ice during the MIS31 interval should have been more abundant compared to current conditions. In the NH, sea-ice cover is substantially reduced by up to 15 % in DJF and by up to 50% in the AST and MIS31 runs in JJA (Table 1). Recall however, that the ICTP-CGM boundary conditions represent the 1072 ka maximum warming period and not the entire MIS31 epoch that extends through 1.08 and 1.05 Ma.*

*In order to provide quantitative comparison between global temperature reconstructions and our modelling results multiple paloeproxies are used (Wet et al. (2016)). Table 2 shows 15 sites distributed in both hemisphere (Fig. 3 supp. material). In the NH largest differences between our modeling results and reconstruction are located in extratropical latitudes, namely at the Lake E and ODP 982 sites (Table 2, (Melles et al. (2012) Lawrence et al. (2009)), where the model is colder than reconstructions with values between 2-3°C. These location are dominated by high seasonality that may not be fully captured in the ICTP-CGCM. Over extratropical regions heat advection embedded in storms has been pointed as an important contributor for defining temperature and weather patterns over those regions (Lehmann and Coumou (2015); Jost et al. (2005); Kageyama and Valdes (2000)). Storm tracks, moreover are tightly connected to the meridional thermal gradient, and over the continent more frequent cold spells in winter are related to low storm track activity (Lehmann and Coumou (2015)), which in lower resolution models such as ICTP-CGCM may hamper a better representation of both the structure and intensity of storms, passing over East Asia and North Atlantic. Subsequently weaker storms can induce lower temperature at the Lake E and ODP 982 site. Elsewhere in the NH differences are smaller than 1°C (Raymo et al. (1996); Li et al. (2011); Herbert et al. (2010a); Naafs et al. (2013)).*

*In the equatorial/tropical region (Herbert et al. (2010b) Li et al. (2011) Dyez and Ravelo (2014) Medina-Elizalde et al. (2008) McClymont and Rosell-Melé (2005) Herbert et al. (2010c) Russon et al. (2011)), the model performs in very good agreement with reconstruction with departures by up to ±1°C. Comparing 849, 847, 846 and 871 ODP sites in the equatorial Pacific (Table 2) the east-west SST gradient may be identified as supported by changes in the zonal circulation. Indeed, the MIS31 climate experiences less vigorous trade winds which is in agreement with Martínez-Garcia et al. (2010). The ODP sites evaluated in the SH also corroborate with previous results showing small differences between model and reconstructions (Tab. 2, Scherer et al. (2008); Voelker et al. (2015); Naish et al. (2009); Russon et al. (2011), McClymont et al. (2005) Crundwell et al. (2008) Martínez-Garcia et al. (2010)). As argued by Wet et al. (2016), conclusions on the absolute temperature values during the MIS31 interval may be treated with caution due to the calibration issues. Changes in MOC and OHT*

There is particular interest in evaluating changes to the MOC associated with warming and/or freshening of the NH high-latitude surface waters due to natural variability (as shown here), and/or including anthropogenic induced-global warming (Rahmstorf et al. (2015). The Atlantic MOC is a key element of the climate system, because it carries a substantial amount of heat poleward, and on long timescales, plays an important role in coupling the SH and NH (Broecker (1998)).

Figure 3a,d shows that the ICTP-CGCM properly reproduces the magnitude of the North Atlantic Deep Water (NADW, 20 Sv) compared to data-based estimates and modeling resultsas previously discussed. The main sites of the NADW formation are also properly located. Analysis of the density contribution in the main sites of the NADW formation demonstrate that thermal changes dominate. Indeed, stronger extratropical winds, as compared to mid-latitudes, increase the vertical air-sea temperature contrast and consequently the ocean-atmosphere heat exchange (Schmitt et al. (1989); Speer and Tziperman (1992)). This leads to stronger convective mixing (Fig. 3a). The CTR shows two regions of density gain (Fig. 3a): the North Atlantic and Icelandic Sea where cold and dry air masses blow over relatively warm water, much weaker activity occurs in the Nordic and Labrador Seas.

The modification of the WAIS leads to slightly reduced (but significant) rate of formation of the NADW as compared to the CTR (Fig. 3e). The weakening of the NADW in the TOPO simulation is associated with reduced heat exchange between the ocean and the atmosphere in the North Atlantic, Labrador Iceland and Norwegian (GIN) Sea due to increased sea-ice (Table 1, Fig. 2d), thereby reducing convective mixing (Fig. 3b). Reduction in surface salinity is also observed in TOPO simulation as compared to CTR (not shown), in particular in the Labrador Sea.

This is also demonstrated by the surface density anomalies, a combination of the thermal and haline density contributions (Speer and Tziperman (1992); Justino et al. (2014)) (Fig. 3b). It can also be argued that intensified intrusion of Antarctic water between 3000-4000 m (Fig. 3e), in the North Atlantic results in increased vertical oceanic stability/stratification hampering oceanic convection. Similar results have been reported for Last Glacial Maximum conditions where colder conditions in the North Atlantic led to a weaker NADW flow (McManus et al. (2004); Peltier and Solheim (2004)). Changes in topography of the WAIS, shown in Figures 2 and 3, have confined impact and therefore AST and MIS 31 show very similar results. Thus we choose to show only results for MIS 31.

*Turning to the AST and MIS31 experiments, Fig. 3 shows that despite reduced sea-ice, there are thermally-increased surface water density in the main sites of deep water formation, particularly in the Labrador and GIN seas (Fig. 3c,f). Thus, the NADW in these experiments is deeper and intensified compared to the control simulation. Intensified MOC and its associated OHT have also been claimed to prevent NH cooling during the MIS11 interval (Dickson et al. (2009)).*

*As shown by the thermal contribution, increases in the MOC (Fig. 3f) are related to intensified westerly atmospheric flow in the northern North Atlantic (Fig. 2d), leading to strong convective mixing. It can also be argued that less intrusion of the Antarctic water in the North Atlantic above 4000 m results in vertical instability favoring oceanic convection (Haupt and Seidov (2012)). Moreover, southward mass transport between 0-1000 m in the NH mid-latitudes is reduced, as shown by negative anomalies, in the AST and MIS31 simulations compared to the CTR (Fig. 3f). An intensified MOC during the MIS31 has also*

*been suggested by paleoreconstructions (Scherer et al. (2008)).*

    *According to Figure 3c, the convection sites in MIS31 have been shifted poleward compared with CTR. Labrador and GIN Seas concentrate the large amount of deep water formation. This occurs as a result of intensified surface wind field over 60°N latitude belt (not shown), and subsequently strengthening of the subtropical gyre northward of its position in the CTR run. The North Atlantic also experiences increased salinity in the MIS31 as compared to CTR which is advected to the*

*Labrador and GIN sas (Fig. 4d).* Changes in atmospheric and oceanic features, such as those discussed above, also produce modifications in OHT (Fig. 4a). The OHT in the Atlantic is mainly driven by the MOC cell, while in the Pacific it is driven by the horizontal wind-driven circulation (Ferrari and Ferreira (2011)). *Figure 4a shows that our modeled OHT is in the range of global observations based on Ganachaud and Wunsch (2003, 2000). It should be considered that a more criterious worldwide evaluation is however not feasible because there are no observations of global OHT as such, but only hydrographical sections*

*along individual latitude belts or indirect estimates from the residually-derived surface fluxes (Trenberth and Caron (2001); Ganachaud and Wunsch (2003)).*

    *Observations exhibit uncertainties in the magnitude values as discussed by Ganachaud and Wunsch (2003), which are larger in the SH tropical region. At some section the error can be as large as 0.55 PW (petawatt) at 4.5°S in the Atlantic and 0.6 PW at 18°S, which represent in some cases more than 60% of the total OHT estimated.*

*Insofar as the ICTP-CGCM is concerned, it underestimates the OHT in the NH due to limitation in the Atlantic Ocean because the OHT in the Pacific matches (Fig. 4a) the values proposed by Ganachaud and Wunsch (2003) and Ganachaud and Wunsch (2000). For the time being we have to cope with uncertainties still present.*

    The OHT in the TOPO changes slightly compared to the CTR. However, in the AST and MIS31 simulations, a clear pattern of increased astronomically-driven northward OHT is present (Fig. 4a).

In mid-latitudes, intensified atmospheric westerly flow in the vicinity of the American continent is simulated in the AST and MIS31 experiments, which contribute to the enhanced OHT via the transport of warmer subtropical water to mid-latitudes (Fig. 4a). As mentioned, southward mass transport between 0-1000 m in the NH mid-latitudes is reduced (Fig. 3f).

    Interestingly, Fig. 4a also shows the dominant contribution from the Indian-Pacific sector to global OHT anomalies, because no differences in OHT in Atlantic is simimulated in MIS31 compared with CTR. This finding is in line with previous results

demonstrating that enhancement of OHT into the Arctic Ocean results in better correspondence between modeling results and

the Lake E reconstruction (Coletti et al. (2015)). Enhanced OHT during the MIS31 is also supported by the ODP site 806 and 849 (**?**).

The present day OHT in the Pacific sector is associated with the subtropical wind-driven circulation in the western Pacific (Kleeman et al. (1999)); however, under MIS31/AST conditions, important contribution may also arises from density changes.

The wind-driven part may be assessed by computing the Sverdrup transport (Eq. 1), defined as:

$$\psi(x) = \frac{1}{\beta \rho} \int_{x_e}^{x} \frac{\partial \tau_x}{\partial y} dx \qquad (1)$$

where $\beta$ is the meridional derivative of the Coriolis parameter, $\rho$ is the mean density of sea water, and $\tau_x$ is the zonal component of the wind stress. The integral is computed from the eastern to the western boundary in the North Pacific using modeled atmospheric wind stress data. The ICTP-CGCM model simulates the Sverdrup transport quite well (not shown)

compared to the magnitude of the Sverdrup transport estimated from the International Comprehensive Ocean-Atmosphere Data Set (ICOADS) (Woodruff et al. (2011)). However, the front that separates the subtropical and the polar gyre is shifted northward in the ICTP-CGCM compared to the ICOADS data.

The Sverdrup transport anomalies between the MIS31 and control experiments (Fig. 4b) show an overall strengthening by up to 20% of the mass transport between 25°N-45°N, as a result of enhanced wind stress curl. This in turn decreases (increases)

the amount of warmer water (surface density) reaching the North Pacific (see negative SST anomalies) and modified the OHT, as discussed below. *The density contribution (Fig. 3a,b,c) shows that the incorporation of the astronomical forcing in the AST/MIS31 experiments led to potential increases in water density in the North Pacific in consonance with increased surface salinity (see Fig. 4d). Moreover, the initial speed up of the subtropical gyre associated with modified mid-latitude westerlies, and the associated heat loss from the ocean to the atmosphere northward of 40°N, dominates the surface density (Fig. 3c).*

*Subsequently, this leads to the formation of the Pacific Meridional Overturning Circulation (PMOC).*

*Reduction in sea-ice cover reduces the density changes in the sea-ice/water interface, but this contribution to the PMOC weakening is marginal and confined to the Arctic region. Additional contribution to the PMOC is provided by increased evaporation in AST/MIS31 runs compared to the precipitation anomalies, which further increase the surface salinity in the North Pacific (Fig. 4d). Figure 5 (supp. material) summarizes the air-sea interaction mechanisms which are involved in the*

*PMOC formation rate.* Evaluating the individual contributions of the wind-driven and thermohaline circulation to North Pacific OHT across 26°N shows that under CTR conditions from surface to 300 m depth, the wind-driven component contributes up to 58% (0.55 PW) of the total OHT (not shown). These values are similar to previous estimates based on observations (Talley (2003)) showing that in subtropics and mid-latitudes, most of the OHT is due to the North Pacific gyre. An additional 42% (0.40 PW) of OHT occurs in the 300-1200 m layer.

For the MIS31 climate, the OHT associated with the wind-driven circulation (0-300 m) at 26°N is less than during CTR and represents 44% of the total (55% in the CTR). This indicates that at this latitude important contribution to OHT is related to the thermohaline circulation (Fig. 4c), below the Ekman layer. In comparison to the CTR simulation, this represents an increase of 16%, from 40% in the CTR to 56% in the MIS31. It should to be mentioned however, that separating the thermohaline and

wind-driven contributions should be interpreted carefully, as the wind-driven density transport partly drives the thermohaline circulation (Talley (2003)).

As shown by vertically integrating the zonal and meridional OHT at basin scale, the contribution of the gyre circulation is dominant in particular between 30-45°N (Supplementary Figs. 2a,c). Under MIS31 conditions, zonally induced OHT is even stronger (Supplementary Fig. 2b), but northward of 45°N the role of the meridional contribution should be taken into account (Supplementary Fig. 2d). Hence, the evaluation of the OHT in a single longitudinal belt does not fully describe the OHT picture insofar as characteristics of large scale domains are needed.

## 4 Concluding Remarks

Despite limitations associated with the atmospheric model component employed in this study (only 8 vertical levels), the findings reasonably match paleoreconstructions in the framework of the Ocean Drilling Program, ANDRILL, and other individual proxy data (Table 2). These modeling results have enormous implications for paleoreconstructions of the MIS31 climate that mostly assume overall ice free conditions in the vicinity of the Antarctic continent. Since these reconstructions may depict dominant signals in a particular time interval and locale, they cannot be assumed to geographically represent large-scale domains and their ability to reproduce long-term environmental conditions should be considered with care. Finally, it is important to emphasize that understanding past interglacial intervals that are characterized by a depleted WAIS can shed light on the potential effects of increasing atmospheric $CO_2$, as the stability of the WAIS will be a key climate factor in decades to come.

*Author contributions.* F. Justino designed the study, wrote large portions of the manuscript and performed, data processing and plotting. D. Lindemann and F. Kucharski performed all model simulations. All authors substantially contributed to interpretation of the results.

*Competing interests.* The authors declare that they have no competing financial interests.

*Acknowledgements.* This work was supported by the the Brazilian National Research Council projects 232718/2014-8 and 407681/2013-2.

5   The first author also thank the Byrd Polar and Climate Research Center for providing the necessary infrastructure.

. NOAA-OI-surface temperature-V2 and ICOADS data are provided by the NOAA/OAR/ESRL PSD, Boulder, Colorado, USA, from their Web site at http://www.esrl.noaa.gov/psd/. ERAI E - P flux was provided by the National Center for Atmospheric Research Staff (Eds),"The Climate Data Guide: ERA-Interim: derived components.", retrieved from https://climatedataguide.ucar.edu/climate-data/era-interim-derived-components, last modified 02 Jan 2014.

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

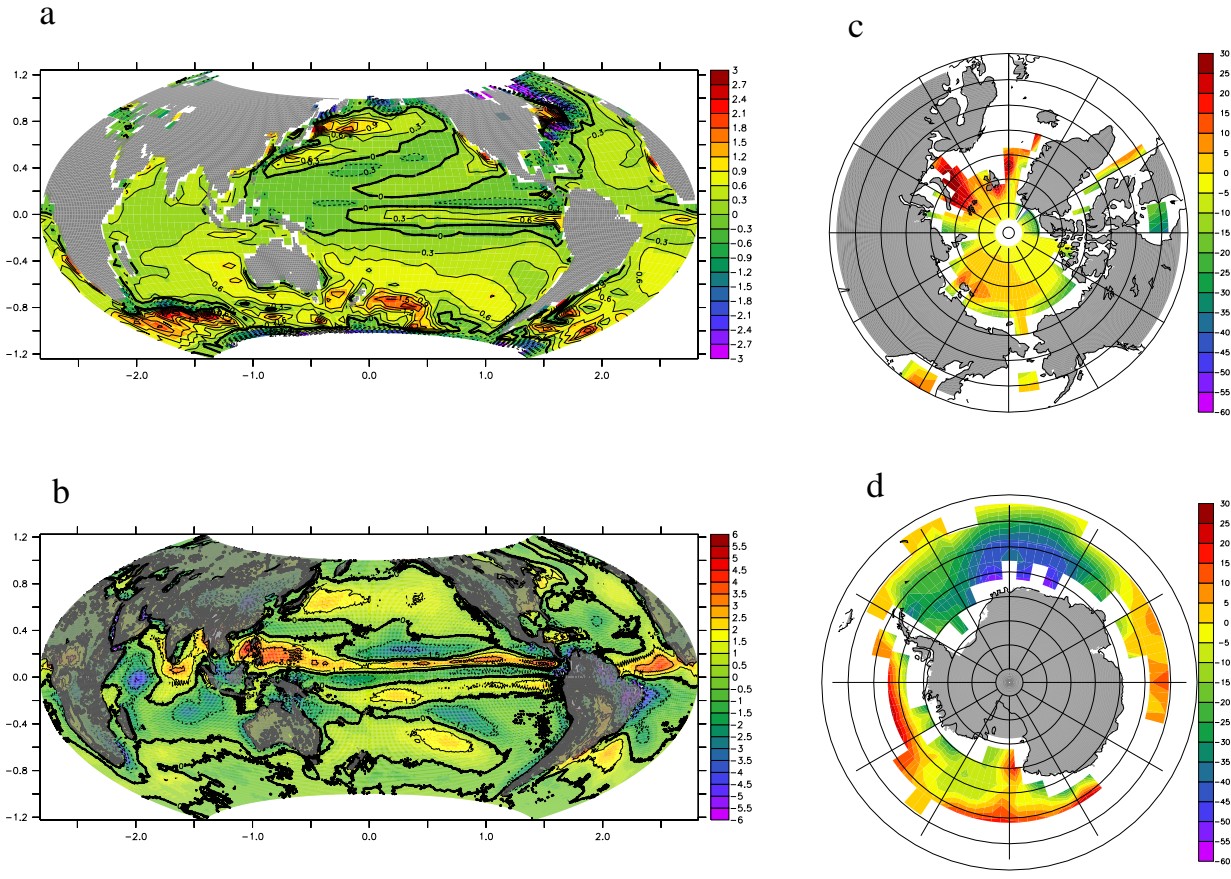

**Figure 1.** (a) Sea surface temperature differences (°C) between the CTR and the NOAA-OI-surface temperature-V2 . (b) E - P flux differences (mm day$^{-1}$) between the control simulation and the ERAI. Sea-ice area differences (%) between the CTR and the HadISST estimates, (c) Northern Hemisphere (d) Southern Hemisphere.

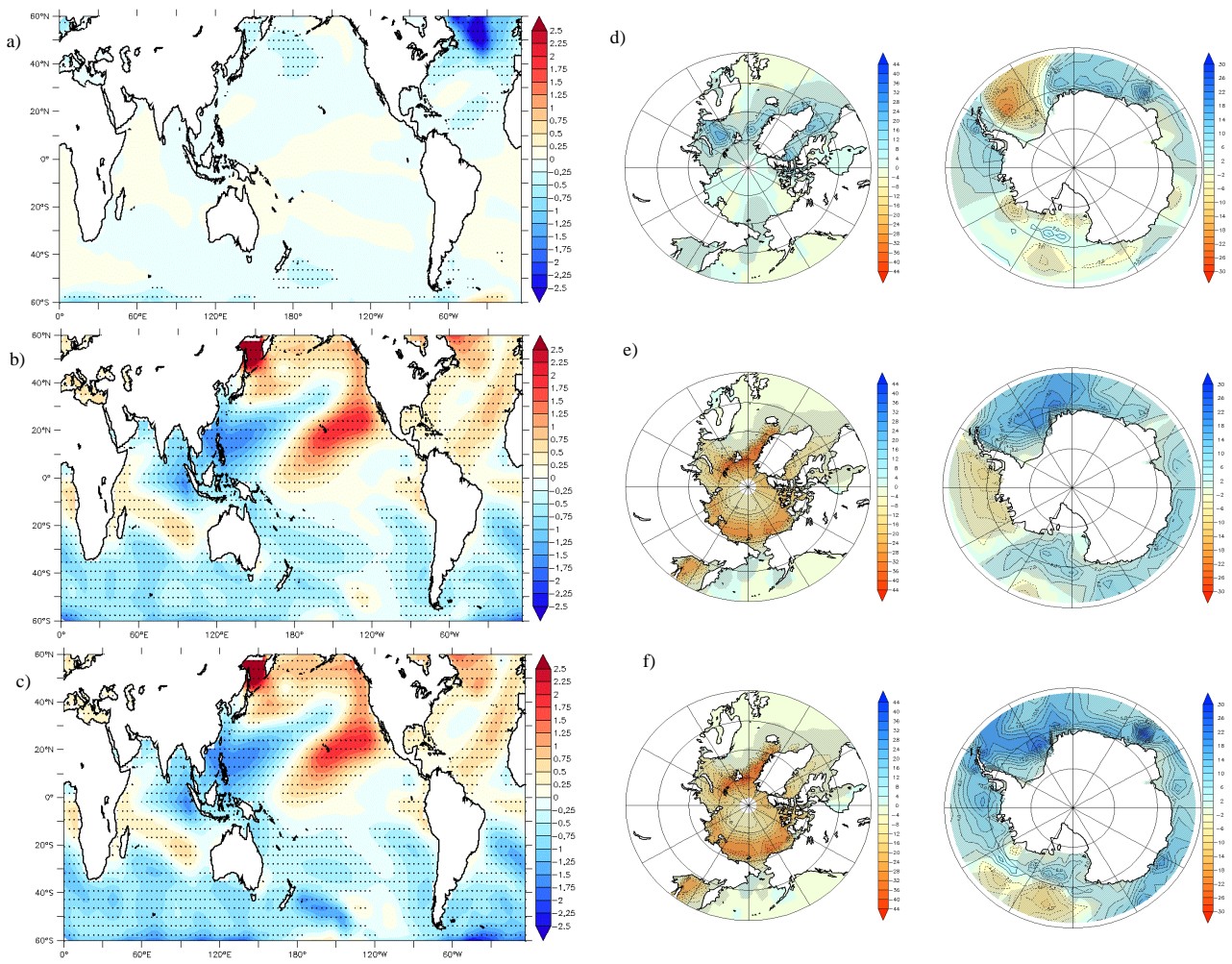

**Figure 2.** Surface temperature differences (°C) between (a) TOPO, (b) AST, and (c) MIS31 compared to the CTR. d), e) and f) are the same as a,b,c but for sea-ice differences (%). Dotted areas in a), b), and c) and hatched in d), e) and f) are significant at 95% based on t-test statistics.

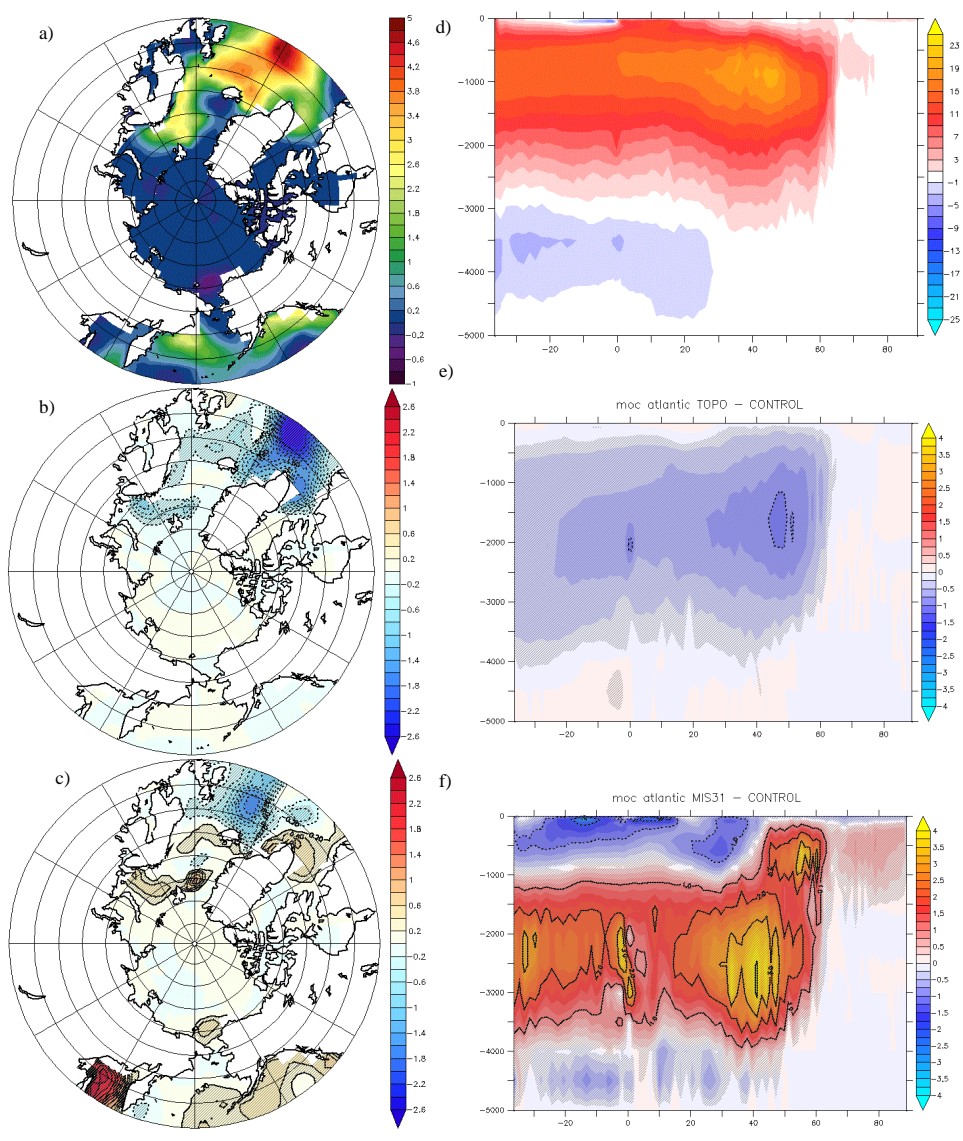

**Figure 3.** Density flux for CTR (a, $10^6 \mathrm{kgm}^{-2}\mathrm{s}^{-1}$) and differences between TOPO - CTR (b) and (c) MIS31 - CTR. (d) MOC (Sv) in the CTR and differences between the (e) TOPO - CTR and (f) MIS31 - CTR. Hatched areas are significant at 95% based on t-test statistics.

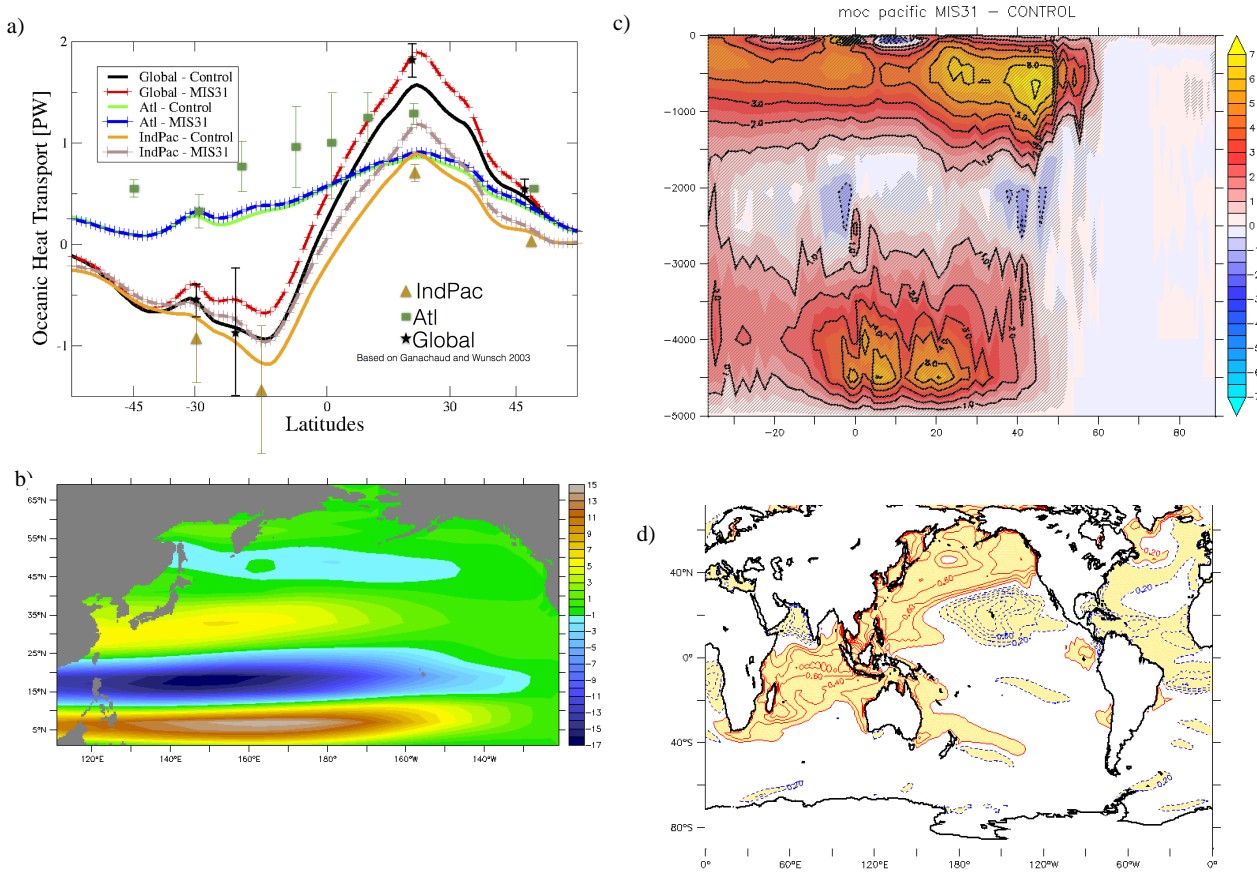

**Figure 4.** (a) Time-avaraged OHT (PW) for CTR (solid line) and MIS31 (dashed-crossed line). Stars, squares and triagles show estimates based on Ganachaud and Wunsch (2003). (b) Sverdrup transport differences (Sv) between the MIS31 and CTR. (c) Differences between the MIS31 and CTR MOC in the Pacific ocean. (d) Sea surface salinity differences between MIS31 and MIS31. Hatched regions in b) and yellow shaded in c) are significant at 95% based on t-test statistics.

**Table 1.** Sea-ice area ($10^9$ m$^2$) in the NH and SH for Hadley Centre Sea Ice (in brackets), CTR and differences between the sensitivity experiments and CTR. Values with star are statistically significant at 95% based on t-test statistics.

|  | NH | | SH | |
|---|---|---|---|---|
|  | DJF | JJA | DJF | JJA |
| CTR (Hadley) | 13.09 (13.36) | 8.23 (8.68) | 4.90 (5.10) | 13.93 (13.08) |
| TOPO-CTR | 0.8 | 0.5 | 0.03 | 0.4 |
| AST-CTR | -1.3* | -4.0* | 0.9 | 1.6 |
| MIS31-CTR | -1.4* | -4.2* | 1.3 | 2.3 |

**Table 2.** SST paleorecontruction and modeling intercomparison. Based on Wet et al. (2016)

| Site (coordinates) | Surf.Temp. ($^o$C) Reconstruction | Surf. Temp. ($^o$C) Speedy-NEMO | Differences between Speedy-NEMO and Reconstructions ($^o$C) | Reference |
|---|---|---|---|---|
| Lake E (67N 172E) | 14.3 | 12.5 | -2.2 | Melles et al. (2012) |
| ODP 982 (57N 15W) | 13.8 | 10.8 | -3.0 | Lawrence et al. (2009) |
| DSDP607 (41N 33W) | 17.5 | 16.9 | 0.6 | Raymo et al. (1996) |
| 306-U1313 (41N 32W ) | 18.0 | 16.9 | -1.1 | Naafs et al. (2013) |
| 1146 (19N 116E) | 26.0 | 25.0 | -1.0 | Herbert et al. (2010a) |
| 722 (16N 59W) | 27.0 | 28.0 | 1.0 | Herbert et al. (2010b) |
| 1143 (9N 113E) | 28.3 | 27.5 | -0.8 | Li et al. (2011) |
| 871 (5N 172E) | 29.3 | 28.9 | -0.4 | Dyez and Ravelo (2014) |
| 847 (0 95W) | 25.6 | 25.0 | -0.6 | Medina-Elizalde et al. (2008) |
| 849 (0 110W) | 25.8 | 25.0 | -0.8 | McClymont and Rosell-Melé (2005) |
| 846 (3S 90W) | 24.3 | 24.8 | 0.5 | Herbert et al. (2010c) |
| MD-06-301 (23S 166E) | 25.0 | 23.9 | -1.1 | Russon et al. (2011) |
| 1087 (31S 15E) | 18.0 | 17.7 | -0.3 | McClymont et al. (2005) |
| 1123 (41S, 171E) | 16.0 | 16.8 | 0.8 | Crundwell et al. (2008) |
| 1090 (42S 8E) | 11.5 | 9.8 | -1.7 | Martínez-Garcia et al. (2010) |