# Peer review of "Oceanic response to changes in the WAIS and astronomical forcing during the MIS31 superinterglacial"

_Climate of the Past, 2016_

## Referee Comment (RC1) · Anonymous Referee #1 · 30 Dec 2016

Review of the manuscript "Oceanic response to changes in the WAIS and astronomical forcing during the MIS31 superinterglacial" by Justino et al.

The manuscript by Justino et al. aims at investigating the response of the ocean and sea ice dynamics due to the exceptional astronomical forcing of MIS 31 as well as due to the grealty reduced elevation of the MIS 31 West Antarctic Ice sheet. To investigate those impacts, Justino et al use the coupled AOGCM ICTP-CGCM at broadly 3° of horizontal resolution. They perform a serie of simulations aiming at isolating the impact of MIS 31 astronomical forcing, of the absence of the WAIS and of a combination of both (most realistic simulation of MIS 31 global climate). Results show that the topographic effect of the absence of the WAIS does not have a significant impact on the ocean heat

transport but affect the extent of the sea ice cover. On the contrary, both the ocean heat transport and the sea ice cover extent are greatly influenced by the astronomical parameters values representative of the warmest epoch during MIS 31.The main impact is that the increased ocean heat transport contributes to a significant reduction of the sea ice extent in the Northern Hemisphere, both in the Atlantic and in the Pacific due to an enhanced meridional ocean transport.

The questions investigated in the manuscript are certainly interesting and timely with the research of Antarctic tipping points under various different climate states. However, to me, the manuscript has been written too fast, many sections are confusing some parts are repetitive and not very well written and structured. I am not a native English speaker my self but it seems to me that the English can be largely improved by the co-authors of this manuscript. I detailed further about the weaknesses of the structure of the manuscript. About the results, most of the Figures have not been done with care, with colorscales that induce lots of confusion during the reading. Most of the signals discussed in this manuscript are small, and no statistics has been performed to understand if most of the signal results significant at 95% (for example with a t-test). In addition, the areas where most the changes occur, so with large anomalies are located in the areas where the model displays the strongest biases for present-day. Moreover, most of the manuscript deals with ocean heat transport, which, the authors admit, is largely biased in their present-day control simulation. I therefore question the impact and significance of the results presented in the manuscript. I list in the following, some suggestions on how to improve the structure, the figures and the interpretation. For specific comments and minors tipoes, I annotated the manuscript directly.

Based on the comments and on the general statement written above, the manuscript is not ready for publication unless a substantial work is done to strengthen the overall results. I recommend major revisions for this manuscript.

***General comments:

- I found the manuscript very superficial in the explanations of the main processes in act. It is very descriptive but never really explain why changes are happening from dynamical point of view between all the experiments. - plot on each Figure when possible, the areas that are statistically significant at 90% and above and only interpret those in the manuscript. - show that the model biases do not significantly impact on the results themselves and their interpretation with supportive material. - The paper would need a proper discussion section to analyse the limitations of this approach and oft he model. - Similarly, the authors should show the impact of the low resolution on location of the deep water convection site,s in particular in the southern ocean (never shown). Because it is well known that some models at low resolution do not capture the weddell sea deep water convection site (and others, like in the Labrador Sea). Very often, most of the meridional circulation happens then in the Pacific sector. - The Figures have to be improved by changing the colorscale, centered the values around zero etc. . . - The use of the literature references is sometimes approximated and sometimes erroneous

****About science: - Perform t-test or other, on all the Figures and Tables (in particular) to understand the real significance of the differences. - Put a figure of oceanic heat transport for the CTR run, in comparison with observations in the supplementary material. - Insert a frame of density changes also for southern ocean in the main manuscript - A figure of stream function showing the deep water formations sites in the Atlantic and in the Pacific sector of the southern Ocean

The rest of my comments have been directly annotated in the pdf of the manuscript in attachment.

Please also note the supplement to this comment:
http://www.clim-past-discuss.net/cp-2016-113/cp-2016-113-RC1-supplement.zip

---

## Referee Comment (RC2) · Anonymous Referee #2 · 2 Feb 2017

This paper explores the impact of changing astronomical forcing and West Antarctic topography on the climate of the MIS31 super interglacial. The study uses a coupled ocean-atmosphere model, with a relatively coarse atmospheric component. This is an improvement on previous studies that have prescribed ocean heat transport with slab ocean models; more could be made of this novelty in the introduction. The main result is that the astronomical configuration for MIS31 contributes to increased northward ocean heat transport (predominantly in the Pacific Ocean), reduced Artic sea ice cover and warmer northern hemisphere temperatures.

MIS31 is an interesting period to study, however the questions that this study is trying to address are not well defined in the introduction. The records from Lake El'gygytgyn

are referenced in the introduction, with the difficulty in simulating MIS31 warmth given as justification for this study. However it is the exceptional warmth of MIS31 relative to other interglacials (MIS1 and 5e) that is of real interest. It would make for a much more satisfying study if simulations for other interglacials were also included (e.g. for MIS1, 5e and 11, following the experimental design of Coletti et al. 2014). The mechanisms discussed in this paper are interesting, but they could also apply to MIS5e. An anomaly plot of simulations for MIS31 and MIS5e would add greatly to this study.

There is no attempt to make a quantitative comparison between the paleorecords and the model output beyond a warmer/cooler comparison (Fig. 2c), the simulations of Melles et al. 2012 and Coletti et al. 2014 have shown that MIS31 was warmer than modern. Statistical tests on the significance of the differences shown would also be useful.

There are some relevant references that could be added: de Wet et al., 2016, EPSL; DeConto et al., 2012, Global Planetary Change; Villa et al., 2012, Global Planetary Change. Additionally there are a number of statements throughout the manuscript that require references.

Overall the manuscript needs major revisions and careful editing of the revised manuscript, as it is quite difficult to follow in its present form, some minor changes are listed below:

Page (line)

2(28): Although it makes sense to use the same atmospheric CO2 for modern and MIS31 for the purposes of this study, there could be some discussion of uncertainty on the MIS31 CO2 estimates and what role this could play in the exceptional warmth of MIS31. 3(12): "It has to be mentioned that", is informal. There are similar statements throughout the manuscript (e.g. 5(29), 9(16)). In most cases these can simply be removed from the beginning of the sentence. 3(25): Given the importance of OHT to the study it would be useful to include a figure with modern day differences and the discuss what impact these biases may have on the results. 5(2): "SPEEDO" has not been defined previously. 9(33): It has not been shown convincingly that there is good agreement between the model output and paleoreconstructions. Need references: 1(22); 2(30); 5(2); 5(4); 10(4).

———————————————————

---

## Author Comment (AC1) · 10 Mar 2017

Flavio Justino

Universidade Federal de Vicosa

PH Rolfs S/N

Vicosa, MG

Brazil

fjustino@ufv.br

To the Editor Climate of the Past (CP):

Our paper, "**Oceanic response to changes in the WAIS and astronomical forcing during the MIS31 superinterglacial**" is reviewed.

Please find enclosed point-by-point replies to the reviewer comments and suggestions. We greatly appreciate all comments and careful evaluation done by the anonymous reviewers, which will substantially improve the manuscript.

Sincerely,

Flavio Justino

**Reviewer #1:**

General comments:

The reviewer points to the relevance of the manuscript (MS) in the framework of Antarctic tipping point, insofar as the melting of West Antarctic Ice Sheet is concerned. This is a point which we had not explicitly mentioned, but it is an interesting one and we will include a statement on this in the revised MS. Moreover, the reviewer finds that modifications in sea-ice cover as discussed in the MS may shed some light on the potential impact of global warming in the extra-tropical latitudes.

The reviewer states, however, that the MS is written too fast and some sections needs better structure. We have modified the MS in this revised version where suggested by the reviewer. The reviewer also suggested co-authors should help improve the English. This will be taken care of.

A major concern of the reviewer is related to the model biases in the extra-tropical latitudes. This is an important point indeed. As discussed in several publications the representation of extra-tropical SST and sea-ice are currently among the largest limitations of Earth climate modeling.

Based on a CORE model intercomparison (Griffies, et al 2009 *Coordinated Ocean-ice Reference Experiment (CORES), 2009,Ocean Modelling, doi:10.1016/j.ocemod.2008.08.007)* we are finding that the Speedy-nemo biases are in the lower range as compared to other models. We will include this reference and a discussion of this point in the revised MS. In this intercomparison with driven ocean only simulations it was demonstrated that even in this idealized scenario generally models appear to have large biases in all fields (e.gSST, SSS, sea ice, zonal velocity in Equatorial Pacific sub-surface, their Fig 7 and 8).In particular, the models' AMOCs (Fig. 23) show substantial spread, but Kiel-ORCA performs very well. Speedy-NEMO, which is the ocean component

used in our study, is a coupled ocean-atmosphere model that applies the same Kiel-ORCA ocean component. It gives us some confidence in our model that Kiel-ORCA is among the better models of the CORE model intercomparison. This is supported by a recent publication by Kucharski et al 2015 (cited in the MS).

As discussed in the MS our AMOC exhibits values that closely match observation, e.g. compared to Ferrari and Ferreira (2011) and Talley (2003) (cited in the MS). In the revised MS we will also point to the fact that higher resolution models with the same base as Speedy_NEMO estimate the AMOC well (Stepanov and Haines 2014 doi:10.5194/os-10-645-2014, Griffies, et al 2009 doi:10.1016/j.ocemod.2008.08.007, Sterl et al 2012 Cli. Dyn.). Consequently, a fair representation of the oceanic heat transport (OHT) should be expected under present day conditions because the majority of the OHT is driven by the AMOC.

The reviewer also argue that we admit that the OHT in our simulation is largely biased. However, this is a misunderstanding because the text in Page 4, Lines 25-30 in the original MS refers to the SPEEDO model instead of Speedy-NEMO. The former has in fact limitation in reproducing the AMOC. The statement was used to justify the use of a more complex ocean model utilized in our MS.

The reviewer suggested the inclusion of a figure showing modeled OHT versus observations. This is however not feasible because there are no observations of global OHT as such, but only hydrographical sections along individual latitude belts or indirect estimates from the residually-derived surface fluxes. Our Figure 4a matches very closely other model estimates and blended data (see Trenberth and Caron 2001, Trenberth and Fasullo 2017) for present day conditions, as shown in the figure below (included here, but not in the revised MS).

[Figure]

**Figure 3.** Northward energy transports: The annual and zonal means of the northward energy transports for 2000–2014 in PW for (left) the total Earth system (black), the atmosphere (red) and the ocean (blue). (right) The ocean component broken down into the contributions from the Atlantic (violet), Pacific (red), and Indian (green) Oceans which combine south of 35°S to give the southern ocean value, as given in the small map below. For dOHC/dt, data only through 2013 are considered. The error bars are ±1 standard deviation.

Therefore, we do not intend to include such a comparison in the revised MS We will mention in the revised MS that the peak of Atlantic OHT varies in position and magnitude among estimates in energy balance approximations and in climate model results.

As suggested by the reviewer, the revised version includes t-test statistics for differences between the CTR simulation and the sensitivity experiments. This is shown at the end of this document.

Comments annotated in the PDF file by the reviewer:

1. PAGE 1. In fact there exists changes of the MOC and OHT in both Atlantic and Pacific, but in the latter they are stronger. This will be modified in the abstract.

2. PAGE 1. We have explained the mechanisms responsible for changes in the PMOC (Figure 5 flowchart in original MS). We do not have reason to believe that these changes are related to the model biases due to its resolution. Speedy-NEMO is run in a reasonable resolution for a global model in particular in the tropics where most of the OHT is transported. The same applies for changes in sea-ice in the sensitivity experiments.

3. PAGE 2. The sentence will be modified to: "Additionally, 325 ppm characterizes the $CO_2$ concentration by the year 1950 which does not include the increase in $CO_2$ due to human emission in the end of the 20th century.

4. PAGE 3 -1. The analyses have been conducted for the last 100 years of a 1000 year -long simulation.

   PAGE 3 -2.  As demonstrated in Fig 4 supp. material, our coupled model is able to reproduce the main sites of deep water formation in the SH.

   PAGE 3 -3, 4. The manuscript focuses on annual mean changes of the MIS31 climate. Discussion of the seasonal cycle, though very important, is out of the scope of the paper.

   PAGE 3 -4, 5. To address the reviewer comment that our vertical wind structure and the jet is shifted southward as compared to those of reanalysis, we show below the zonal wind profile.

[Figure]

[Figure]

Here it is seen that despite limitation in our atmospheric component of the coupled model, Speedy is suitable for our study. Additional analyses are provided in http://users.ictp.it/~kucharsk/speedy8_clim.html.

PAGE 3 -6. We will include in the conclusion limitations of our analyses as well as caveats related to the modeling framework.

PAGE 4 -1. Statement will be modified according to reviewer suggestion.

PAGE4 -2. No, we have not included changes in the initial salinity field in response to the WAIS collapse. This has been treated similarly previously in Justino et al (2014 Cli. Dyn.). This is supported by Aiken and England (2008) who demonstrated limited response of the climate system to the freshening implied by Antarctic sea ice melt.

Moreover, Vaughan and Spouge (2001) argued that an outflow rate associated with WAIS melting is not realistically attainable, making it difficult to implement in a rose experiment. However, changes in temperature around Antarctica might be expected by adding freshwater. This has been included in the revised MS.

The MS focuses on analyzing the climate response to changes in the Antarctic topography due to WAIS collapse, insofar mechanical changes in orography lead to modified atmospheric lapse-rate.

PAGE 4 -3. All figures will be modified accordingly

PAGE 4 -4. The sentence will be removed.

PAGE 4 -5. Yes, Speedy-NEMO can properly capture the sites of deep water formation in the Northern Hemisphere as shown in Figure 3a. This is also true in the Southern Hemisphere as shown in the supplementary material Figure 4. This will be pointed out in the revised MS.

PAGE 4 -6. The statement will be removed as suggested to Section 2.1.

PAGE 5 -1. References will be added to observations and modeling based studies (Stepanov and Haines 2014 doi:10.5194/os-10-645-2014, Griffies, et al 2009 doi:10.1016/j.ocemod.2008.08.007, Sterl et al 2012 Cli. Dyn.)

PAGE 5 -2. Reference will be included, Mathiot et al. 2010
http://dx.doi.org/10.1016/j.ocemod.2010.07.001

PAGE 5 -3,4 Brackets will be included and Figure 2 will be modified

PAGE 5 -5. In conditions of reduced sea-ice thickness there is an increase in the heat flux from the ocean to the atmosphere further increasing the convective mixing. The exchange of heat and mass between the atmosphere and ocean is strongly modulated by sea ice and vice-versa.

PAGE 5 -6. We are aware that seasonal analyses of the MIS31 sea-ice characteristics are important for understanding the global climate. However, in our analyses of MIS31, our main focus is climatic features that vary on long time-scales, such as AMOC, PMOC and OHT. A thorough discussion of seasonal changes is beyond the scope. We will explain this shortly.

PAGE 5 -7. Reference to Yin and Berger (2012) will be added

PAGE 5 -8. Paragraph will be removed

PAGE 6 -1. Surface temperature refers to SST or land surface temperature. This will be clarified.

PAGE 6 -2. This statement is important because it emphasizes the astronomically driven air-sea interaction which is a crucial mechanism related to changes in SST. The reviewer has requested to include temperature values in the dots and squares shown in Figure 2c. This is not practical because the original reconstructions exhibit large uncertainties in the equatorial temperatures, as they been inferred from changes in the Walker circulation. The same applies for the polar region though the inference for cooling/warming is based on different processes. Wet et al 2016, EPSL argued that "we hesitate to draw conclusions on the absolute temperature values reached during the studied interval due to the calibration issues, numerous interesting features are apparent based on relative temperature changes." Highlighting the complexity of proxy-model data intercomparison

PAGE 6 -4. The statement will be re-phrased.

PAGE 6 -5. The reviewer argues that the model bias can limit the reliability of our findings. It is well known that all coupled models exhibit limitations in particular over the polar regions, as assessed by the IPCC AR5 (shown in the figure below).

Evaluation of sea ice in models is hampered by insufficient observations of some key variables (e.g. ice thickness). Nevertheless, particular climate anomalies resulting from inclusion of distinct boundary conditions may be primarily assumed to be climate-driven. Though, we will emphasize in the revised version limitations in our simulation of sea–ice in the Weddell Sea.

PAGE 6 -6. We compared the sea-ice extent in all experiments MIS31,TOPO and AST. This is important to provide to the reader an evaluation of the individual impacts of implementing the boundary conditions. Moreover, this can shed light on non-linear effects of the joint forcing (TOPO + AST) applied in the MIS31 run.

[Figure]

(Top and middle rows) Time series of sea ice extent from 1900 to 2012 for (a) the Arctic in September and (b) the Antarctic in February, as modelled in CMIP5 (coloured lines) and observations-based (NASA; Comiso and Nishio, 2008) and NSIDC; (Fetterer et al., 2002), solid and dashed thick black lines, respectively). The CMIP5 multi- model ensemble mean (thick red line) is based on 37 CMIP5 models (historical simulations extended after 2005 with RCP4.5 projections). Each model is represented with a single simulation. The dotted black line for the Arctic in (a) relates to the pre-satellite period of observation-based time series (Stroeve et al., 2012). In (a) and (b) the panels on the right are based on the corresponding 37-member ensemble means from CMIP5 (thick red lines) and 12-model ensemble means from CMIP3 (thick blue lines). The CMIP3 12-model means are based on CMIP3 historical simulations extended after 1999 with Special Report on Emission Scenarios (SRES) A2 projections. The pink and light blue shadings denote the 5 to 95 percentile range for the corresponding ensembles. Note that these are monthly means, not yearly minima. (Adapted from Pavlova et al., 2011.) (Bottom row) CMIP5 sea ice extent trend distributions over the period 1979–2010 for (c) the Arctic in September and (d) the Antarctic in February. Altogether 66 realizations are shown from 26 different models (historical simulations extended after 2005 with RCP4.5 projections). They are compared against the observations-based estimates of the trends (green vertical lines in (c) and (d) from Comiso and Nishio (2008); blue vertical line in (d) from Parkinson and Cavalieri (2012)). In (c), the observations-based estimates (Cavalieri and Parkinson, 2012; Comiso and Nishio, 2008) coincide.

PAGE 7 -1, 2, 3. We will provide new figures, but kept the subsection "*Changes in MOC and OHT.*" Reference will be included (Stouffer et al 2007).

PAGE 7 -4. Stronger mean winds refer to comparison to annual mean conditions, this occurs for instance in winter months. This may also apply for the sensitivity experiments in comparison to CTR simulation. This will be better explained.

PAGE 7 -5,6. New figure and the statistical significance of differences will be provided for Table 1.

PAGE 7 -7. The reviewer is right, there is no clear evidence indicating a shallower cell in the TOPO. The statement will be removed.

PAGE 7 -8,9,10. The paragraph is modified.

PAGE 7 -11. We will implement in the revised MS as suggested: "changes in topography of the WAIS, shown in Figures 2 and 3, have no significant impact and therefore AST and MIS 31 show very similar results. Thus we choose to show only results for MIS 31."

PAGE 7 -12. The reviewer is right, we have not discussed changes in the main site of NADW formation between CTR and MIS31. To clarify this we will include in the revised MS: "The joint effect of the astronomical and WAIS topography forcings in the MIS31 climate is to increase density flux in the Labrador Sea and the North Atlantic in the MIS31, as compared to the CTR counterpart (Figure 3c). Another source of NADW formation during the MIS31 interglacial is located in the Norwegian Sea, as shown in Figure 3f."

PAGE 7 -13. All figures have been redone including t-test statistics. This has shown that our statement on the intrusion of AABW in the North Atlantic included in the original MS is valid.

PAGE 14 -1. Statement will be removed.

PAGE 14 -2. In fact, superficial transport does not decrease. In CTR simulation the zonal mean flow in the North Atlantic is southward between 20N-Equator (Fig. 3d) whereas in MIS31 it shifts northward with maximum between 20N-40N. This will be clarified in the revised MS.

PAGE 14 -3. The reviewer suggestion will be included.

PAGE 14 -4 Figure will be included in the supplementary material.

PAGE 14 -5. This paragraph shows the initial mechanisms related to the formation of the PMOC. The flowchart (supp. Material Fig 4) explains in more detail the climate interaction related to the PMAC formation.

PAGE 14 -6. Paragraph will be modified to include the reviewer suggestion.

**Reviewer** #2

The main comment raised by the reviewer concerns the possibility of comparing our MIS31 simulation with similar experiments of MIS1 and 5e. We recognize that seeing our MIS31 experiments in relation to these two other interglacials would add to the manuscript value. However, this will require another set of experiments specifically 6 additional runs. We regret that at this stage is not feasible to proceed as suggested by the reviewer, as new modeling experiments could not be conducted in due time. Because an AOGCM is used, demanding computational time and complexity in interpreting global results make this task un-attainable. In fact, it is for the first time that such experiments have been performed with a full rather than a slab oceasn model. We will leave this interesting comparison to a potential follow up publication. However, all other comments by this reviewer are addressed.

We will modify the introductory section to better define the manuscript focus. Also we will emphasize clearer that our study is an improvement of previous ones conducted with slab ocean models. Indeed, this is the first study conducted with an AOGCM to evaluate the

MIS31 interglacial, performed to disentangle individual climate responses to astronomical and WAIS topography forcings.

We will add the suggested references, and their main findings in the Introduction.

We will include a paragraph on the CO2 uncertainties during MIS31, and their potential impact on our results which assume present day CO2.

There are unfortunately no observations of global OHT as presented in the MS, we have therefore, compared with indirect estimations. This point was raised also by reviewer #1, and in our reply to her/him we explain in some more detail why we will refrain from including such a comparison.

Regarding the paleo-model inter-comparison shown in Figure 2c, we will argue that the MIS31 interval lacks extensive reconstructions, and those available do not provide magnitudes, but rather in general express whether the climate state was cooler or warmer than the present climate. This is our reason for showing red squares (warming) and blue squares (cooling) together with modeled temperature anomalies.

* Regarding to $CO_2$, $CH_4$ and $N_2O$ concentration.

It has been proposed by Hoenisch et al. (2009) that the MIS31 has the highest partial pressure of $CO_2$ of the mid-Pleistocene, by about 325 ppm. However, according to their Figure 1, the CO2 concentration could vary between 300 and 350 ppm during the MIS31, due to propagated error of the individual pH, SST, salinity, and alkalinity. The uncertainty in the atmospheric composition may lead to overestimation in the NH warming as simulated in our study. Changes in $CO_2$ by about +50 ppm may be associated with +0.3K change in globally averaged surface temperature. In fact, this alteration in temperature is within the uncertainties of the climate sensitivity (Bindoff et al. 2013). The $CH_4$ (800 ppb, Loulergue et al. 2008) and $N_2O$ (288 ppb, Schilt et al. 2010) concentrations are similar to Coletii et al. (2015).

[Figure]

(a) Surface temperature differences (!C) between the CTR and the NOAA-OI-surface temperature-V2 . The white shading indicates surface temperatures -1.8!C. (b) Sea-ice cover in the CTR (shaded in %) and the sea ice (yellow line)based on HadISST. (c) Time-averaged E - P flux differences (mm day−1) between the control simulation and the ERAI.

[Figure]

Figure 2. Surface temperature differences (C) between (a) TOPO, (b) AST, and (c) MIS31 compared to the CTR. Sea-ice differences (%) between the runs (d, e, f) Land-ocean reconstructions are shown as red squares (warmer MIS31 conditions) and blue squares (colder MIS31 conditions) as compared to CTR simulation. Dotted areas are significant at 95% based on t-test statistics.

[Figure]

.
Figure 3. Density flux for CTR (a, $10^{-6}$ kg m$^{-2}$ s$^{-1}$) and differences between the sensitivity experiments and CTR (b) TOPO, (c) MIS31.(d) Time-averaged MOC (Sv) in the CTR and differences between the CTR and (e) TOPO and (f) MIS31. Hatched areas are significant at 95% based on t-test statistics.

[Figure]

Figure 4. (a) OHT (PW) for CTR (solid line) and MIS31 (dashed-crossed line). (b) Sverdrup transport differences (Sv) between the MIS31 and CTR. (c) Differences between the MIS31 and CTR MOC in the Pacific ocean (shaded, Sv), and contour shows the Pacific MOC in CTR. (d) Surface salinity differences between MIS31 and CTR. Hatched (Yellow) areas are significant at 95% based on t-test statistics in c (d).

---

## Author Response (AR1)

Flavio Justino

Universidade Federal de Viçosa

PH Rolfs S/N

Vicosa, MG

Brazil

fjustino@ufv.br

To the Editor Climate of the Past (CP):

Our paper, "**Oceanic response to changes in the WAIS and astronomical forcing during the MIS31 superinterglacial**" is reviewed.

Please find enclosed replies to the reviewer comments and suggestions. We greatly appreciate all comments and careful evaluation done by the anonymous reviewers, which substantially improved the manuscript. In the revised MS italic parts are completely modified. Light comments and types have also been included as suggested.

Sincerely,
Flavio Justino

A major concern of the reviewers is related to the model biases in the extra-tropical latitudes, as well as associated with the Meridional Overturning Circulation (MOC) and Oceanic Heat Transport (OHT) magnitudes. This is an important point indeed, and to provide the reader with comparison of our modeled values, the revised MS includes in figure 4 observations based on Ganachaud and Wunsch (2000; 2003) for OHT and Talley (2003), Griffies, et al (2009), Sterl et al (2012), Stepanov and Haines (2014) for the MOC. An extensive discussion of the model caveats is included in the MS pages 3-4, insofar as SST, sea-ice and E-P flux are concerned. The CTR climate simulation is compared to HadSSTI dataset for SST and sea-ice and to ERAI for E-P flux.

Reviewer 2 also suggested an intercomparison of our MIS31 simulation with similar experiments for the MIS1 and 5e interglacial. We recognize that seeing our MIS31 experiments in relation to these two other intervals would be very useful. However, this will require another set of experiments specifically 6 additional runs. We regret that at this stage is not feasible to proceed as suggested by the reviewer, as new modeling experiments could not be conducted in due time. Because an AOGCM is used, demanding computational time and complexity in interpreting global results make this task un-attainable in due time. In fact, it is for the first time that such experiments have been performed with a full rather than a slab ocean model. We will leave this interesting comparison to a potential follow up publication.

The introductory section has been substantially modified to better define the manuscript focus as suggested. Also it is emphasized that our study is an improvement of previous ones conducted with slab ocean models. Indeed, this is the first study conducted with an AOGCM to evaluate the MIS31 interglacial, performed to disentangle individual climate responses to astronomical and WAIS topography forcings.

Regarding the paleo-model inter-comparison shown in Figure 2c of the original MS version, we are confortable in saying that the revised MS includes an extensive inter-comparison between modeled results and 19 globally distributed proxies, as shown in the figure below.

[Figure]

This Figure is included in the Supplementary Material. Moreover, in revised MS the Table 2 shows temperature values based on paleo-proxies and differences from the MIS31 simulation (see below). This approach is based on results by Wet et al (2016, EPSL). The discussion is provided in pages 8-9.

| Site (coordinates) | Surf.Temp.($^0$C) reconstruction | Surf. Temp. ($^0$C) Speedy-NEMO | Differences between Speedy-NEMO and reconstructions ($^0$C) |
|---|---|---|---|
| Lake E (67N 172E) | 14.3 | 12.5 | -2.2 |
| ODP 982 (57N 15W) | 13.8 | 10.8 | -3.0 |
| DSDP607 (41N 33W) | 17.5 | 16.9 | 0.6 |
| 306-U1313 (41N 32W ) | 18.0 | 16.9 | -1.1 |
| 1146 (19N 116E) | 26.0 | 25.0 | -1.0 |
| 722 (16N 59W) | 27.0 | 28.0 | 1.0 |
| 1143 (9N 113E) | 28.3 | 27.5 | -0.8 |
| 871 (5N 172E) | 29.3 | 28.9 | -0.4 |
| 847 (0 95W) | 25.6 | 25.0 | -0.6 |
| 849 (0 110W) | 25.8 | 25.0 | -0.8 |
| 846 (3S 90W) | 24.3 | 24.8 | 0.5 |
| MD-06-301 (23S 166 E) | 25.0 | 23.9 | -1.1 |
| 1087 (31S 15E) | 18.0 | 17.7 | -0.3 |
| 1123 (41S, 171E) | 16.0 | 16.8 | 0.8 |
| 1090 (42S 8E) | 11.5 | 9.8 | -1.7 |

Regarding uncertainties in $CO_2$ concentration during the MIS31, we argued that Hoenisch et al. (2009) proposes that the MIS31 has the highest partial pressure of $CO_2$ of the mid-Pleistocene, by about 325 ppm. However, according to their Figure 1, the $CO_2$ concentration could vary between 300 and 350 ppm during the MIS31, due to propagated error of the individual pH, SST, salinity, and alkalinity. This uncertainty in the atmospheric composition may lead to overestimation in the NH warming as simulated in our study. Changes in $CO_2$ by about +50 ppm are however associated with only +0.3K change in globally averaged surface temperature. This alteration in temperature is within the uncertainties of the climate sensitivity (Bindoff et al. 2013), and therefore may not strongly compromise our results.

It has to be stressed that all figures in the revised MS include statistical significance based on t-test statistics.

REVIEWER #1

Comments annotated in the PDF file by the reviewer:

1. PAGE 1. In fact there exists changes of the MOC and OHT in both Atlantic and Pacific, but in the latter they are stronger. This will be modified in the abstract.
2. PAGE 1. We have explained the mechanisms responsible for changes in the PMOC (Figure 5 flowchart in original MS). We do not have reason to believe that these changes are related to the model biases due to its resolution. Speedy-NEMO is run in a reasonable resolution for a global model in particular in the tropics where most of the OHT is transported. The same applies for changes in sea-ice in the sensitivity experiments.
3. PAGE 2. The sentence will be modified to: "Additionally, 325 ppm characterizes the CO2 concentration by the year 1950 which does not include the increase in CO2 due to human emission in the end of the 20th century.
4. PAGE 3 -1. The analyses have been conducted for the last 100 years of a 2000 year -long simulation.

PAGE 3 -3, 4. The manuscript focuses on annual mean changes of the MIS31 climate. Discussion of the seasonal cycle, though very important, is out of the scope of the paper.

PAGE 3 -4, 5. To address the reviewer comment that our vertical wind structure and the jet is shifted southward as compared to those of reanalysis, we show below the zonal wind profile.

[Figure]

[Figure]

Here it is seen that despite limitation in our atmospheric component of the coupled model, Speedy is suitable for our study. Additional analyses are provided in http://users.ictp.it/~kucharsk/speedy8_clim.html.

PAGE 3 -6. We have discussed throughout the revised MS limitations of our analyses as well as caveats related to the modeling framework.

PAGE 4 -1. Statement will be modified according to reviewer suggestion.

PAGE4 -2. No, we have not included changes in the initial salinity field in response to the WAIS collapse. This has been treated similarly previously in Justino et al (2014 Cli. Dyn.). This is supported by Aiken and England (2008) who demonstrated limited response of the climate system to the freshening implied by Antarctic sea ice melt.

Moreover, Vaughan and Spouge (2001) argued that an outflow rate associated with WAIS melting is not realistically attainable, making it difficult to implement in a rose experiment. However, changes in temperature around Antarctica might be expected by adding freshwater. This has been included in the revised MS.

The MS focuses on analyzing the climate response to changes in the Antarctic topography due to WAIS collapse, insofar mechanical changes in orography lead to modified atmospheric lapse-rate.

PAGE 4 -3. All figures is modified accordingly

PAGE 4 -4. The sentence is removed.

PAGE 4 -5. Yes, Speedy-NEMO can properly capture the sites of deep water formation in the Northern Hemisphere as shown in Figure 3a. This is pointed out in the revised MS.

PAGE 4 -6. The statement will be removed as suggested to Section 2.1.

PAGE 5 -1. References will be added to observations and modeling based studies (Stepanov and Haines 2014 doi:10.5194/os-10-645-2014, Griffies, et al 2009 doi:10.1016/j.ocemod.2008.08.007, Sterl et al 2012 Cli. Dyn.)

PAGE 5 -2. Reference will be included, Mathiot et al. 2010
http://dx.doi.org/10.1016/j.ocemod.2010.07.001

PAGE 5 -3,4 Brackets will be included and Figure 2 is modified

PAGE 5 -5. In conditions of reduced sea-ice thickness there is an increase in the heat flux from the ocean to the atmosphere further increasing the convective mixing. The exchange of heat and mass between the atmosphere and ocean is strongly modulated by sea ice and vice-versa.

PAGE 5 -6. We are aware that seasonal analyses of the MIS31 sea-ice characteristics are important for understanding the global climate. However, in our analyses of MIS31, our main focus is climatic features that vary on long time-scales, such as AMOC, PMOC and OHT. We have shortly discussed in the revised MS changes in sea-ice.

PAGE 5 -7. Reference to Yin and Berger (2012) will be added

PAGE 5 -8. Paragraph is removed

PAGE 6 -1. Surface temperature refers to SST or land surface temperature. This is clarified.

PAGE 6 -2. This statement is important because it emphasizes the astronomically driven air-sea interaction which is a crucial mechanism related to changes in SST. We have included the Table 2 showing paleo-modeling intercomparison.

PAGE 6 -4. The statement is re-phrased.

PAGE 6 -5. The reviewer argues that the model bias can limit the reliability of our findings. It is well known that all coupled models exhibit limitations in particular over the polar regions, as assessed by the IPCC AR5.

Evaluation of sea ice in models is hampered by insufficient observations of some key variables (e.g. ice thickness). Nevertheless, particular climate anomalies resulting from inclusion of distinct boundary conditions may be primarily assumed to be climate-driven. Though, we will emphasize in the revised version limitations in our simulation of sea–ice in the Weddell Sea.

PAGE 6 -6. We compared the sea-ice extent in all experiments MIS31,TOPO and AST. This is important to provide to the reader an evaluation of the individual impacts of implementing the boundary conditions. Moreover, this can shed light on non-linear effects of the joint forcing (TOPO + AST) applied in the MIS31 run.

PAGE 7 -1, 2, 3. We provided new figures, but kept the subsection "*Changes in MOC and OHT.*" Reference will be included (Stouffer et al 2007).

PAGE 7 -4. Stronger mean winds refer to comparison to annual mean conditions, this occurs for instance in winter months. This may also apply for the sensitivity experiments in comparison to CTR simulation. This will be better explained.

PAGE 7 -5,6. New figure and the statistical significance of differences is provided for Table 1.

PAGE 7 -7. The reviewer is right, there is no clear evidence indicating a shallower cell in the TOPO. The statement is removed.

PAGE 7 -8,9,10. The paragraph is modified.

PAGE 7 -11. We implemented in the revised MS as suggested: "changes in topography of the WAIS, shown in Figures 2 and 3, have no significant impact and therefore AST and MIS 31 show very similar results. Thus we choose to show only results for MIS 31."

PAGE 7 -12. The reviewer is right, we have not discussed changes in the main site of NADW formation between CTR and MIS31. To clarify this we will include in the revised MS: "The joint effect of the astronomical and WAIS topography forcings in the MIS31 climate is to increase density flux in the Labrador Sea and the North Atlantic in the MIS31,

as compared to the CTR counterpart (Figure 3c). Another source of NADW formation during the MIS31 interglacial is located in the Norwegian Sea, as shown in Figure 3f."

PAGE 7 -13. All figures have been redone including t-test statistics. This has shown that our statement on the intrusion of AABW in the North Atlantic included in the original MS is valid.

PAGE 14 -1. Statement will be removed.

PAGE 14 -2. In fact, superficial transport does not decrease. In CTR simulation the zonal mean flow in the North Atlantic is southward between 20N-Equator (Fig. 3d) whereas in MIS31 it shifts northward with maximum between 20N-40N. This is clarified in the revised MS.

PAGE 14 -3. The reviewer suggestion is included.

PAGE 14 -5. This paragraph shows the initial mechanisms related to the formation of the PMOC. The flowchart (supp. Material Fig 4) explains in more detail the climate interaction related to the PMOC formation.

PAGE 14 -6. Paragraph is modified to include the reviewer suggestion.

**Reviewer** #2

The main comment raised by the reviewer concerns the possibility of comparing our MIS31 simulation with similar experiments of MIS1 and 5e. We recognize that seeing our MIS31 experiments in relation to these two other interglacials would add to the manuscript value. However, this will require another set of experiments specifically 6 additional runs. We regret that at this stage is not feasible to proceed as suggested by the reviewer, as new modeling experiments could not be conducted in due time. Because an AOGCM is used, demanding computational time and complexity in interpreting global results make this task un-attainable. In fact, it is for the first time that such experiments have been performed with a full rather than a slab oceasn model. We will leave this interesting comparison to a

potential follow up publication. However, all other comments by this reviewer are addressed.

We modified the introductory section to better define the manuscript focus. Also we will emphasize clearer that our study is an improvement of previous ones conducted with slab ocean models. Indeed, this is the first study conducted with an AOGCM to evaluate the MIS31 interglacial, performed to disentangle individual climate responses to astronomical and WAIS topography forcings.

We added the suggested references, and their main findings in the Introduction.

We included a paragraph on the CO2 uncertainties during MIS31, and their potential impact on our results which assume present day CO2.

* Regarding to $CO_2$, $CH_4$ and $N_2O$ concentration.

It has been proposed by Hoenisch et al. (2009) that the MIS31 has the highest partial pressure of $CO_2$ of the mid-Pleistocene, by about 325 ppm. However, according to their Figure 1, the CO2 concentration could vary between 300 and 350 ppm during the MIS31, due to propagated error of the individual pH, SST, salinity, and alkalinity. The uncertainty in the atmospheric composition may lead to overestimation in the NH warming as simulated in our study. Changes in $CO_2$ by about +50 ppm may be associated with +0.3K change in globally averaged surface temperature. In fact, this alteration in temperature is within the uncertainties of the climate sensitivity (Bindoff et al. 2013). The $CH_4$ (800 ppb, Loulergue et al. 2008) and $N_2O$ (288 ppb, Schilt et al. 2010) concentrations are similar to Coletii et al. (2015).

---

## Author Response (AR2)

Flavio Justino

Universidade Federal de Viçosa

PH Rolfs S/N

Vicosa, MG

Brazil

fjustino@ufv.br

To the Editor

Climate of the Past (CP):

Our paper, "**Oceanic response to changes in the WAIS and astronomical forcing during the MIS31 superinterglacial**" is reviewed.

Please find enclosed replies to the comments and suggestions. We greatly appreciate all comments and careful evaluation done by the Editor, which substantially improved the manuscript. In the revised MS, *italic* parts stand for modified paragraphs and news statements. Lighter comments that do not lead to misinterpretation of results have been removed. Typos have also been corrected but now always edited in *italic*. All figures have been modified to highlight differences that are statistically significant.

The 3$^{rd}$ version of our Manuscript has been carefully revised by all co-authors including Aaron Wilson and David Bromwich, both native English speaker.

In the following we have sequentially addressed all points raised, as shown below.

Sincerely,

Flavio Justino

**Abstract:**

The word "substantial" has been REMOVED

**Introduction:**

-Reference/citation style has been revised according to the CP LaTeX template.

PAGE 2

-The expression "such as" has been REMOVED and citation added

- Citation added (Coletti et al 2015, Lund et al 2017, Roychowdhury and DeConto, 2016)

-Sentence removed *"This approach is explored here by using a coupled model"*

-Sentence removed *"At large …"*

-Sentence moved to Page 2, line 16 as suggested.

PAGE 3

-Sentence moved to Section 3, Page 4 line 8 as suggested.

-Section 2 has been modified starting with a description of the model characteristics.

-Removed all informal text such as "*It has to be mentioned …*
"

**EDITOR REMARKS PAGE 4:** Given the reviewers concerns about potential bias, it is critical that you are honest about areas where the model performs less well too. Will this difference not have influence on your deep-water formation locations and magnitude?

Editor comments on page 6 most deal with biases in deep-water formation.

In order to address this issue we have compared our simulated AABW and NADW with oceanographic data as suggested. This is included in the revised MS as the initial part of the subsection "Changes in MOC and OHT" PAGES 8 and 9:

*The ICTP-CGCM AMOC exhibits values that closely match observations (Kanzow et al., 2010; Ferrari and Ferreira, 2011; Talley et al., 2003) as well as higher*

resolution models (Stepanov and Haines, 2014). Consequently, a fair representation of the AMOC should lead to proper OHT estimates under present day conditions, because the majority of the OHT is driven by the AMOC. Moreover, the resolution of ICTP-CGCM across the tropics is sufficient enough to capture the majority of the OHT for the globe.

The Antarctic bottom-water (AABW) is closely related to sea-ice processes that involve brine release due to sea-ice formation and winds (Stössel et al., 1998). Despite limitations in reproducing the sea-ice seasonal features in the ICTP-CGCM, simulated deep water formation in the SH occurs in both the Atlantic and Pacific Oceans (Figs. 3-4).

The AABW represented by our CTR run in the Atlantic attains values of about 5 Sv ($10^6$ $m^3$ $s^{-1}$), which is comparable to 8 Sv based on absolute geostrophic velocity from hydrographic data (Talley et al., 2003) and from climatological Ekman transports. At 25°S, the ICTP-CGCM delivers AABW in the Pacific Ocean up to 10 Sv also in line with Talley et al. (2003), and 6-8 Sv at 10°N matched values found by Wijffels et al. (1996).

Figures 3a and 3d show that compared to data-based estimates (Ganachaud and Wunsch, 2000), the ICTP-CGCM properly reproduces the magnitude of the North Atlantic Deep Water (NADW, 15 ± 2 Sv) at 24°N. The main sites of the NADW formation, namely Greenland-Iceland, Norwegian (GIN) and Labrador Seas (Wood et al., 1999) are also properly located as shown by analyses of the density contribution (Fig. 3a). Thermal changes dominate the NADW formation in comparison to the haline contribution. Indeed, a much colder extra-tropical atmosphere over the warmer ocean increases the vertical air-sea temperature contrast and consequently the ocean-atmosphere heat exchange (Schmitt et al., 1989; Speer and Tziperman, 1992). This leads to stronger convective mixing (Fig. 3a).

-Citation removed: Kucharski et al, 2015.

PAGE 5.

-MIS 31 included as suggested: applies the MIS 31 WAIS topography...
-References included: *Dahl-Jensen et al., (2013), and Coletti et al., (2015)*

PAGE 6 Lines 1-5. The revised version includes Figures in global projection including the polar region.
-We have clarified in the text that our comparison is to the CTR run.

PAGE 6, Lines 6-11. Modified as suggested (Page 6).

PAGE 7. McCreary and Lu 1994.

PAGE 7-8. We have clarified the discussion on the joint effect of WAIS topography and astronomical forcing in leading the MIS31 climate, as follow:

*The global climate response due to the combined effect of changing WAIS topography and astronomical forcing (MIS31 simulation) is primarily a result of changes in the latter forcing, as Fig. 2c shows a similar SST anomaly pattern as Fig. 2b. Nevertheless, the combined forcing appears not to be linear in the vicinity of Antarctica (Supplementary Fig. 2). Linearity is noted, however, by intensified warming in the Ross Sea as a result of warmer SSTs in TOPO and AST compared to the CTR climate. Non-linearity is shown through reduced cooling in the Weddell Sea in the MIS31 simulation compared to the AST simulation (Supplementary Fig. 2). This is related to the absence of the WAIS topography that reduces the strong cooling associated with changes in the astronomical forcing.  Comparison between the MIS31 and the AST runs can be indirectly used to further identify the effect of the WAIS topography in the SH sea-ice changes (Figs. 2e and 2f).*

PAGE 8, 1st paragraph: Figures 2a,b,c in the revised version include the polar regions

PAGE8, 2nd paragraph: The sentence is included as an open statement to highlight the influence of air-sea coupling in leading sea-ice anomalies between the MIS31 forcing and the CTR climate. It has been added to the revised version:

*Modification of the WAIS topography is associated with changes in sea-ice area, particularly in the Atlantic Ocean. Changes in the astronomical forcing on the other hand are more responsible for climate anomalies on a global perspective. Differences between MIS31 and AST usefully demonstrate that the substantial reduction of sea-ice cover in the Ross Sea and in some extent changes in Weddell Sea are substantially affected by the WAIS collapse (Supplementary Fig. 1c). Specifically, the MIS31 simulation is warmer in the Weddell and Ross Seas by up to 1.5°C with respect to AST, which is accompanied by an approximately 10% reduction in sea-ice cover. In fact, the individual influence of the collapse of WAIS in MIS31 is more evident in the Bellingshausen Sea (Figs. 2e and 2f). In the NH, the removal of WAIS and orbital forcing act in opposite directions for sea-ice changes.*
*The sensitivity experiments demonstrate that compared to CTR, warmer SSTs and reduced sea-ice are only simulated in the Ross Sea region. This is in agreement with the Cape Roberts Project-1 results and data from the Antarctic Geological Drilling project (ANDRILL) (Naish et al., 2009) (Fig. 2). In fact, outside of the Ross Sea, Antarctic sea-ice during the MIS31 interval may have been more abundant compared to current conditions. In the NH, sea-ice cover is substantially reduced b*

PAGE 8 – ORB and Weddell have been removed
PAGE 8 Line 21 and 9 Line 10,: Sentence removed.

PAGE 9 and 10. We have a provided better description and references that confirm

reasonable representation of the MOC in our CTR climate. The revised version includes in our view a solid explanation for changes in the MOC during the MIS31 epoch. It should be noted that we have provided new Figures (3 and 4). See revised MS pages 9-10 *in italic*.

**PAGE 10, line 25 - Editor comment On the Oceanic Heat Transport (OHT)**

R. Our assumption that weaker OHT in the Northern Hemisphere is due to limitation in the Atlantic Ocean, is related to the fact that the OHT in the Pacific is in the range of proposed values, as shown in Figure 4a (yellow triangles). By evaluating the modeled values in the Atlantic is clear that they reach up to 0.6 PW, whereas "observation" estimates assume values as high as 1 +- 0.3 PW (green squares, Figure 4a). Thus, the global projection, which is the sum of all ocean basins contribution will result in model underestimation due to lower Atlantic OHT.

PAGE 11. Citation added (ODP site 846 Herbert et al., (2010c) and 849 McClymont and Rosell-Melé, (2005)

PAGE 11 on the Sverdrup transport:
We have removed parts involving the Sverdrup transport in the current revised version.

A section **"Summary and Concluding Remarks"** has been modified to include our main findings and shed some light on the need for a better coverage of paleo-proxies in the SH polar Ocean, because at present, they may not hemispherically represent the most dominant characteristics of the polar sea surface in the MIS31 interval, insofar as sea-ice is concerned.

---

## Author Response (AR3)

Flavio Justino

Universidade Federal de Viçosa

PH Rolfs S/N

Vicosa, MG

Brazil

fjustino@ufv.br

To the Editor
Climate of the Past (CP):

Our paper, "**Oceanic response to changes in the WAIS and astronomical forcing during the MIS31 superinterglacial**" is reviewed.

Please find enclosed replies to the comments and suggestions.

**EDITOR -** To proceed, please upload your final version of the manuscript, addressing the following technical corrections:

- remove the italicised sections where you have highlighted your responses to previous comments

*Modified*

- page 4 line 13: can you insert a reference or two here to support your statement about some models being warmer than others?

*Included:* Dawson, A., Matthews, A. J., Stevens, D. P., Roberts, M. J., and Vidale, P. L.: Importance of oceanic resolution and mean state on the extra-tropical response to El Nino in a matrix of coupled models, Climate dynamics, 41, 1439–1452, 2013.

- title of section 3 "Design of the sensitivity experiments"
*Modified*

- delete page 5 lines 13-14 "Our simulation... MIS 31 interval" (you essentially said it on line 6, and it interrupts your discussion of freshwater impacts)

*Modified*

- page 7 line 16-17 notes impact of WAIS topography on sea ice, but this seems to interrupt the overall discussion on SSTs, whereas lines 24-25 then note the impact on sea ice. Should the first of these be moved to coincide or to replace the second? I'm a bit confused about why you cite Supp Fig 1c in the second example when the first cites Figs 2e and 2f?
- page 7 final line (into page 8): is this finding from your results, or do you need to cite one of the data papers which shows this?

*Modified*

- page 8 line 9 "these locations" - clarify if this is Lake E and ODP 982, or remove the paragraph break so that the two sentences are together.

*Paragraph removed*